# Releasing Graph Neural Networks with Differential Privacy Guarantees

**Iyiola E. Olatunji**                                                                 *iyiola@l3s.de*
*L3S Research Center*
*Germany*

**Thorben Funke**                                                                      *tfunke@l3s.de*
*L3S Research Center,*
*Germany*

**Megha Khosla**                                                                   *m.khosla@tudelft.nl*
*TU Delft,*
*Netherlands*

**Reviewed on OpenReview:** *https: // openreview. net/ forum? id= wk8oXR0kFA*

## Abstract

With the increasing popularity of graph neural networks (GNNs) in several sensitive applications like healthcare and medicine, concerns have been raised over the privacy aspects of trained GNNs. More notably, GNNs are vulnerable to privacy attacks, such as membership inference attacks, even if only black-box access to the trained model is granted. We propose PrivGnn, a privacy-preserving framework for releasing GNN models in a centralized setting. Assuming an access to a public unlabeled graph, PrivGnn provides a framework to release GNN models trained explicitly on public data along with knowledge obtained from the private data in a privacy preserving manner. PrivGnn combines the knowledge-distillation framework with the two noise mechanisms, random subsampling, and noisy labeling, to ensure rigorous privacy guarantees. We theoretically analyze our approach in the Rènyi differential privacy framework. Besides, we show the solid experimental performance of our method compared to several baselines adapted for graph-structured data. Our code is available at `https://github.com/iyempissy/privGnn`.

## 1 Introduction

In the past few years, graph neural networks (GNNs) have gained much attention due to their superior performance in a wide range of applications, such as social networks Hamilton et al. (2017), biology Ktena et al. (2018), medicine Ahmedt-Aristizabal et al. (2021), and recommender systems Fan et al. (2019); Zheng et al. (2021). Specifically, GNNs achieve state-of-the-art results in various graph-based learning tasks, such as node classification, link prediction, and community detection.

Real-world graphs, such as medical and economic networks, are associated with sensitive information about individuals and their activities. Hence, they cannot always be made public. Releasing models trained on such data provides an opportunity for using private knowledge beyond company boundaries Ahmedt-Aristizabal et al. (2021). For instance, a social networking company might want to release a content labelling model trained on the private user data without endangering the privacy of the participants Morrow et al. (2022). However, recent works have shown that GNNs are vulnerable to membership inference attacks Olatunji et al. (2021); He et al. (2021); Duddu et al. (2020). Specifically, membership inference attacks aim to identify which data points have been used for training the model. The higher vulnerability of GNNs to such attacks as compared to traditional models has been attributed to their encoding of the graph structure within the

model Olatunji et al. (2021). In addition, the current legal data protection policies (e.g. GDPR) highlight a compelling need to develop *privacy-preserving GNNs.*

We propose our framework PRIVGNN, which builds on the rigid guarantees of *differential privacy* (DP), allowing us to protect sensitive data while releasing the trained GNN model. DP is one of the most popular approaches for releasing data statistics or trained models while concealing the information about individuals present in the dataset Dwork et al. (2006). The key idea of DP is that if we query a dataset containing $N$ individuals, the query's result will be almost indistinguishable (in a probabilistic sense) from the result of querying a neighboring dataset with one less or one more individual. Hence, each individual's privacy is guaranteed with a specific probability. Such probabilistic indistinguishability is usually achieved by incorporating sufficient noise into the query result.

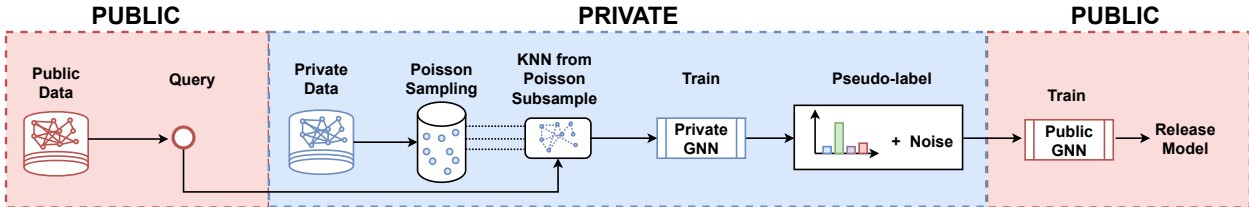

Figure 1: Workflow of PRIVGNN. Given *labeled private* and *unlabeled public* datasets, PRIVGNN starts by retrieving a subset of the private data using Poisson sampling. We then obtain the K-nearest neighbor nodes based on the features of the public query node. The teacher GNN model is trained on the graph induced on K-nearest neighbors. We obtain a pseudo-label for the query node by adding independent noise to the output posterior. The pseudo-label and data from the public graph are used in training the student model, which is then released.

The seminal work of Abadi et al. (2016) proposed *differential private stochastic gradient descent* (DP-SGD) algorithm to achieve differential privacy guarantees for deep learning models. Specifically, in each training step, DP-SGD adds appropriate noise to the $\ell_2$-clipped gradients during the stochastic gradient descent optimization. The incurred privacy budget $\varepsilon$ for training is computed using the moment's accountant technique. This technique keeps track of the privacy loss across multiple invocations of the noise addition mechanism applied to random subsets of the input dataset.

Besides the slow training process of DP-SGD, the injected noise is proportional to the number of training epochs, which further degrades performance. While DP-SGD is designed for independent and identically distributed data (i.i.d.), the nodes in graph data are related. In fact, GNNs make explicit use of the relational structure by recursive feature propagation over connected nodes. Hence, the privacy guarantee of DP-SGD, which requires a set of i.i.d. examples to form batches (subset of the training data that is used in a single iteration of the model training process) and lots (group of one or more batches), does not trivially hold for GNNs and graph data Igamberdiev & Habernal (2021).

To work around DP-SGD's dependency on the training procedure, such as the number of epochs, Papernot et al. (2016) proposed *Private Aggregation of Teacher Ensembles* (PATE). PATE leverages a large ensemble of teacher models trained on disjoint subsets of private data to transfer knowledge to a student model, which is then released with privacy guarantees. However, splitting graph data into multiple disjoint training sets destroys the structural information and adversely affects accuracy.

Since existing DP methods are not directly applicable to GNNs, we propose a privacy-preserving framework, PRIVGNN, for releasing GNN models with differential privacy guarantees. Similar to PATE's dataset assumption, we are given two graphs: a labeled private graph and an unlabeled public graph. PRIVGNN leverages the paradigm of knowledge distillation. The knowledge of the teacher model trained on the private graph is transferred to the student model trained only on the public graph in a differential privacy manner. PRIVGNN achieves practical privacy guarantees by combining the student-teacher training with two noise mechanisms: **random subsampling** using Poisson sampling and **noisy labeling mechanism** to obtain pseudo-labels for public nodes. In particular, we release the student GNN model, which is trained using a small number of public nodes labeled using the teacher GNN models developed exclusively for each public

query node. We present a *Rènyi differential privacy* (RDP) analysis of our approach and provide tight bounds on incurred privacy budget or privacy loss. Figure 1 shows an overview of our PRIVGNN approach.

To summarize, **our key contributions** include: (i) a *novel privacy-preserving framework* for releasing GNN models (ii) *theoretical analysis* of the framework using the RDP framework (iii) *large scale empirical analysis* establishing the practical utility of the approach. Besides, we show the robustness of our model against two membership inference attacks.

## 2 Related Works

Graph neural networks (GNNs) Kipf & Welling (2017); Hamilton et al. (2017); Veličković et al. (2018), mainly popularized by graph convolution networks (GCNs) and their variants, compute node representations by recursive aggregation and transformation of feature representations of its neighbors. While they encode the graph directly into the model via the aggregation scheme, GNNs are highly vulnerable to membership inference attacks Olatunji et al. (2021), thus highlighting the need to develop privacy-preserving GNNs.

Existing works on privacy-preserving GNNs mainly focused on a distributed setting in which the node feature values or/and labels are assumed to be private and distributed among multiple distrusting parties (Sajadmanesh & Gatica-Perez, 2020; Zhou et al., 2020; Wu et al., 2021; Shan et al., 2021). Sajadmanesh & Gatica-Perez (2020) assumed that only the node features are sensitive while the graph structure is publicly available. They developed a local differential privacy mechanism to tackle the problem of node-level privacy.

Wu et al. (2021) proposed a federated framework for privacy-preserving GNN-based recommendation systems while focusing on the task of subgraph level federated learning. Zhou et al. (2020) proposed a privacy-preserving GNN learning paradigm for node classification tasks among distrusting parties such that each party has access to the same set of nodes. However, the features and edges are split among them. Shan et al. (2021) proposed a server-aided privacy-preserving GNN for the node-level task on a horizontally partitioned cross-silo scenario via a secure pooling aggregation mechanism. They assumed that all data holders have the same feature domains and edge types but differ in nodes, edges, and labels.

In the centralized setting, Sajadmanesh et al. (2022) obtained low-dimensional node features independent of the graph structure and preserve the privacy of the node features by applying DP-SGD for node-level privacy. These features are then combined with multi-hop aggregated node embeddings, to which Gaussian noise is added. Igamberdiev & Habernal (2021) adapted differentially-private gradient-based training, DP-SGD Abadi et al. (2016), to GCNs for natural language processing task. However, as noted by the authors, the privacy guarantees of their approach might not hold because DP-SGD requires a set of i.i.d. examples to form batches and lots to distribute the noise effectively, whereas, in GNNs, the nodes exchange information via the message passing framework during training. Other works that utilized DP-SGD in computing their privacy guarantees include Daigavane et al. (2021); Zhang et al. (2022). In another line of work, Tran et al. (2022) directly perturbs the node features and graph structure using randomized response.

We provide a comprehensive comparison between PRIVGNN and existing works, outlining the specific differences in Appendix E.

## 3 Our Approach

**Setting and notations.** Here, we follow the setting of Papernot et al. (2016) in which, in addition to the private graph (with node features and labels), a public graph (with node features) exists and is available to us. This is a fair assumption, as even for sensitive medical datasets, there exist some publicly available datasets. We also note that companies usually have huge amounts of data but might only release a subset of it for research purposes. A few examples include item-user graphs in recommender systems like the Movielens GroupLensResearch (2021) and Netflix dataset. Furthermore, in the field of information retrieval, Microsoft has released the ORCAS dataset Craswell et al. (2020), which is a document-query graph that contains click-through data (for a practical real-world example, please refer to Appendix D.3). It is important to note that the nodes of the public graph are unlabeled.

We denote a graph by $G = (V, E)$ where $V$ is the node-set, and $E$ represents the edges among the nodes. Let $X$ denote the feature matrix for the node-set $V$, such that $X(i)$ corresponds to the feature vector for node $i$. We use additionally the superscript $\dagger$ to denote private data, such as the private graph $G^\dagger = (V^\dagger, E^\dagger)$ with node feature matrix $X^\dagger$ and labels $Y^\dagger$. For simplicity of notation, we denote public elements without any superscript, such as the public (student) GNN $\Phi$ or the set of query nodes $Q \subset V$. In addition, we denote a node $v \in V$ and the $\ell$-hop neighborhood $\mathcal{N}^\ell(v)$ with $\mathcal{N}^0(v) = \{v\}$ which is recursively defined as: $\mathcal{N}^\ell(v) = \{u|(u, w) \in E \text{ and } w \in \mathcal{N}^{\ell-1}(v)\}$.

## 3.1 PRIVGNN Framework

We organize our proposed PRIVGNN method into three phases: *private data selection, retrieval of noisy pseudo-labels, and student model training.* The pseudocode for our PRIVGNN method is provided in Algorithm 1. We start by randomly sampling a set of query nodes $Q$ from the public dataset (line 1). We will use the set $Q$ together with the noisy labels (extracted from the private data) for training the student GNN as described below.

---

**Algorithm 1:** PRIVGNN

---

**Input:** Private graph $G^\dagger = (V^\dagger, E^\dagger)$ with private node feature matrix $X^\dagger$ and private labels $Y^\dagger$; unlabeled public graph $G = (V, E)$ with public features $X$

**Hyperparam.:** $K$ the number of training data points for training $\Phi^\dagger$; $\beta$ the scale of Laplacian noise; $\gamma$ the subsampling ratio

**Output:** Student GNN, $\Phi$

**1** Sample public query node-set $Q \subset V$

**2 for** *query* $v \in Q$ **do**

**3**     Select random subset, $\hat{V}^\dagger \subset V^\dagger$, using Poisson sampling:

      For $\sigma_i \sim Ber(\gamma), v_i \in V^\dagger$ select $\hat{V}^\dagger = \{v_i|\sigma_i = 1, v_i \in V^\dagger\}$

**4**     Retrieve the K-nearest-neighbors of $v$ in $\hat{V}^\dagger$; $V^\dagger_{\text{KNN}}(v) = \text{argmin}_K\{\text{d}(X(v), X^\dagger(u)), u \in \hat{V}^\dagger\}$

**5**     Construct the induced subgraph $H$ of the node-set $V^\dagger_{\text{KNN}}(v)$ from the graph $G^\dagger$

**6**     Initialize and train the GNN $\Phi^\dagger$ on subgraph $H$ using the private labels $Y^\dagger_{|H}$

**7**     Compute pseudo-label $\tilde{y}_v$ using noisy posterior of $\Phi^\dagger$:

      $\tilde{y}_v = \text{argmax}\{\Phi^\dagger(v) + \{\eta_1, \eta_2 \ldots, \eta_c\}\}, \eta_i \sim \text{Lap}(0, \beta)$

**8** Train GNN $\Phi$ on $G$ using the training set $Q$ with the pseudo-labels $\tilde{Y} = \{\tilde{y}_v|v \in Q\}$

**9 return** *Trained student GNN model* $\Phi$

---

### 3.1.1 Private data selection (Lines 4-5)

Corresponding to a query, we first obtain a random subsample of the private data using the Poisson subsampling mechanism with sampling ratio $\gamma$ defined as follows (see Def. 1). We denote the retrieved subset of private nodes by $\hat{V}^\dagger$. We will see in our privacy analysis that such a data sampling mechanism leads to amplification in privacy.

**Definition 1 (PoissonSample).** *Given a dataset $X$, the mechanism PoissonSample outputs a subset of the data $\{x_i|\sigma_i = 1, i \in [n]\}$ by sampling $\sigma_i \sim Ber(\gamma)$ independently for $i = 1, \ldots, n$.*

The mechanism is equivalent to the "sampling without replacement" scheme which involves finding a random subset of size $m$ at random with $m \sim \text{Binomial}(\gamma, n)$. As $n \to \infty$, $\gamma \to 0$ while $\gamma n \to \zeta$, the Binomial distribution converges to a Poisson distribution with parameter $\zeta$. In Poisson sampling, each data point is selected independently with a certain probability, determined by the sampling rate $\gamma$. This method ensures that the overall privacy budget is conserved and allows us to more precisely characterize the RDP of our method (See Appendix B).

**Generating query-specific private subgraph.** Our next step is to generate a private subgraph to train a query-specific teacher GNN. We will then use the trained GNN to generate a pseudo-label for the

corresponding query. To generate a query-specific subgraph, we extract the $K$-nearest neighbors of the query node from the retrieved subset $\hat{V}^\dagger$. In other words, for a query node $v \in Q$, we retrieve the node-set $V^\dagger_{\text{KNN}(v)}$ with:

$$V^\dagger_{\text{KNN}}(v) = \text{argmin}_K\{\text{d}(X(v), X^\dagger(u)), u \in \hat{V}^\dagger\},$$

where $\text{d}(\cdot, \cdot)$ denotes a distance function, such as Euclidean distance or cosine distance, and $X(u)$ is the feature vector of the node $u$. The subgraph $H$ induced by the node-set $V^\dagger_{\text{KNN}}(v)$ of the private graph $G^\dagger$ with the subset of features $X^\dagger_{|H}$ and the subset of labels $Y^\dagger_{|H}$ constitute the selected private data (for the query node $v$). We use this selected private data to train a query-specific GNN.

### 3.1.2 Retrieving noisy pseudo-labels (Lines 6-7)

As the next step, we want to retrieve a pseudo-label for the public query node $v$. We train the private teacher GNN $\Phi^\dagger$ using the selected private data: the subgraph $H$ with features $X^\dagger_{|H}$ and labels $Y^\dagger_{|H}$. Then, we retrieve the prediction $\Phi^\dagger(v)$ using the $\ell$-hop neighborhood $\mathcal{N}^\ell(v)$ (corresponding to $\ell$-layer GNN) of the query node $v$ and the respective subset of feature vectors from the public data. To preserve privacy, we add independent Laplacian noise to each coordinate of the posterior with noise scale $\beta = 1/\lambda$ to obtain the noisy pseudo-label $\tilde{y}_v$ of the query node $v$

$$\tilde{y}_v = \text{argmax}\left\{\Phi^\dagger(v) + \{\eta_1, \eta_2 \ldots, \eta_c\}\right\}, \eta_i \sim \text{Lap}(0, \beta).$$

We note that the teacher GNN is applied in an inductive setting. That is, to infer labels on a public query node that was not seen during training. Therefore, the GNN model most effective in an inductive setting like GraphSAGE Hamilton et al. (2017) should be used as the teacher GNN.

### 3.1.3 Student model (transductive) training

Our private models $\Phi^\dagger$ only answer the selected number of queries from the public graph. This is because each time a model $\Phi^\dagger$ is queried, it utilizes the model trained on private data, which will lead to increased privacy costs. The privately (pseudo-)labeled query nodes and the unlabeled public data are used to train a student model $\Phi$ in a transductive setting. We then release the public student model $\Phi$ and note that the public model $\Phi$ is differentially private. The privacy guarantee is a result of the two applied noise mechanisms, which lead to the plausible deniability of the retrieved pseudo-labels and by the post-processing property of differential privacy.

### 3.1.4 Privacy analysis

We start by describing DP and the related privacy mechanism, followed by a more generalized notion of DP, i.e., RDP. Specifically, we utilize the bound on subsampled RDP and the advanced composition theorem for RDP to derive practical privacy guarantees for our approach. For a complete derivation of our privacy guarantee, see Appendix B.

**Differential privacy Dwork et al. (2006).** DP is the most common notion of privacy for algorithms on statistical databases. Informally, DP bounds the change in the output distribution of a mechanism when there is a small change in its input. Concretely, $\varepsilon$-DP puts a multiplicative upper bound on the worst-case change in output distribution when the input differs by exactly one data point.

**Definition 2 ($\varepsilon$-DP ).** *A mechanism $\mathcal{M}: \mathcal{X} \to \Theta$ is $\varepsilon$-DP if for every pair of neighboring datasets $X, X' \in \mathcal{X}$, and every possible (measurable) output set $E \subseteq \Theta$ the following inequality holds:*

$$Pr[\mathcal{M}(X) \in E] \leq e^\varepsilon Pr[\mathcal{M}(X') \in E].$$

**Differential privacy for graph datasets.** In this work, we consider the case where the adjacent datasets are represented by neighboring graphs, namely $G$ and $G'$. One can then consider node-DP in which $G$ and $G'$ differ by a single node and its corresponding adjacent edges or edge-DP in which the neighboring graphs

differ by a single edge. In this paper, we provide guarantees for node-DP which also provides a stronger give a stronger privacy protection than edge differential privacy.

An example of an $\varepsilon$-DP algorithm is the **Laplace mechanism** which allows releasing a noisy output to an arbitrary query with values in $\mathbb{R}^n$. The mechanism is defined as

$$\mathbb{L}_\varepsilon f(x) \triangleq f(x) + \mathrm{Lap}(0, \Delta_1/\varepsilon),$$

where Lap is the Laplace distribution and $\Delta_1$ is the $\ell_1$ sensitivity of the query $f$. Since GNN $\Phi^\dagger$ returns a probability vector, the $\ell_1$ sensitivity is 1. Here the second parameter, $\Delta_1/\varepsilon$ is also referred to as the scale parameter of the Laplacian distribution. Our final guarantees are expressed using $(\varepsilon, \delta)$-DP which is a relaxation of $\varepsilon$-DP and is defined as follows.

**Definition 3 ($(\varepsilon, \delta)$-DP ).** *A mechanism $\mathcal{M}\colon \mathcal{X} \to \Theta$ is $(\varepsilon, \delta)$-DP if for every pair of neighboring datasets $X, X' \in \mathcal{X}$, and every possible (measurable) output set $E \subseteq \Theta$ the following inequality holds:*

$$Pr[\mathcal{M}(X) \in E] \leq e^\varepsilon Pr[\mathcal{M}(X') \in E] + \delta.$$

**Rènyi differential privacy (RDP).** RDP is a generalization of DP based on the concept of Rènyi divergence. We use the RDP framework for our analysis as it is well-suited for expressing guarantees of the composition of heterogeneous mechanisms, especially those applied to data subsamples.

**Definition 4 (RDP Mironov (2017)).** *A mechanism $\mathcal{M}$ is $(\alpha, \varepsilon)$-RDP with order $\alpha \in (1, \infty)$ if for all neighboring datasets $X, X'$ the following holds*

$$D_\alpha(\mathcal{M}(X) || \mathcal{M}(X')) = \frac{1}{\alpha - 1} \log E_{\theta \sim \mathcal{X}} \left[ \left( \frac{p\mathcal{M}(X)(\theta)}{p\mathcal{M}(X')(\theta)} \right)^\alpha \right] \leq \varepsilon.$$

As $\alpha \to \infty$, RDP converges to the pure $\varepsilon$-DP. In its functional form, we denote $\varepsilon_\mathcal{M}(\alpha)$ as the RDP $\varepsilon$ of $\mathcal{M}$ at order $\alpha$. The function $\varepsilon_\mathcal{M}(\cdot)$ provides a clear characterization of the privacy guarantee associated with $\mathcal{M}$. In this work, we will use the following RDP formula corresponding to the Laplacian mechanism

$$\varepsilon_{\mathrm{LAP}}(\alpha) = \frac{1}{\alpha - 1} \log \left( \left( \frac{\alpha}{2\alpha - 1} \right) e^{\frac{\alpha - 1}{\beta}} + \left( \frac{\alpha - 1}{2\alpha - 1} \right) e^{\frac{-\alpha}{\beta}} \right)$$

for $\alpha > 1$, where $\beta$ is the scale parameter of the Laplace distribution. More generally, we will use the following result to convert RDP to the standard $(\varepsilon, \delta)$-DP for any $\delta > 0$.

**Lemma 5 (From RDP to $(\varepsilon, \delta)$-DP).** *If a mechanism $\mathcal{M}_1$ satisfies $(\alpha, \varepsilon)$-RDP, then $\mathcal{M}_1$ also satisfies $(\varepsilon + \frac{\log 1/\delta}{\alpha - 1}, \delta)$-DP for any $\delta \in (0, 1)$.*

For a composed mechanism $\mathcal{M} = (\mathcal{M}_1, \ldots, \mathcal{M}_t)$, we employ Lemma 5 to provide an $(\varepsilon, \delta)$-DP guarantee

$$\delta \Rightarrow \varepsilon : \varepsilon(\delta) = \min_{\alpha > 1} \frac{\log(1/\delta)}{\alpha - 1} + \varepsilon_\mathcal{M}(\alpha - 1), \tag{1}$$

Another notable advantage of RDP over $(\varepsilon, \delta)$-DP is that it composes very naturally.

**Lemma 6 (Composition with RDP).** *Let $\mathcal{M} = (\mathcal{M}_1, \ldots, \mathcal{M}_t)$ be mechanisms where $\mathcal{M}_i$ can potentially depend on the outputs of $\mathcal{M}_1, \ldots, \mathcal{M}_{i-1}$. Then $\mathcal{M}$ obeys RDP with $\varepsilon_M(\cdot) = \sum_{i=1}^t \varepsilon_{M_i}(\cdot)$.*

Lemma 6 implies that the privacy loss by the composition of two mechanisms $\mathcal{M}_1$ and $\mathcal{M}_2$ is

$$\varepsilon_{\mathcal{M}_1 \times \mathcal{M}_2}(\cdot) = [\varepsilon_{\mathcal{M}_1} + \varepsilon_{\mathcal{M}_2}](\cdot).$$

**Privacy amplification by subsampling.** A commonly used approach in privacy is *subsampling* in which the DP mechanism is applied to the randomly selected sample from the data. Subsampling offers a stronger privacy guarantee in that the one data point that differs between two neighboring datasets has a decreased probability of appearing in the smaller sample. See Appendix B, Theorem 8 for details.

## 3.2 Privacy Guarantees of PRIVGNN

**Theorem 7.** *For any $\delta > 0$, Algorithm 1 is $(\varepsilon, \delta)$-DP with*

$$\varepsilon \leq \log\left(\frac{1}{\sqrt{\delta}}\right) + |Q|\log\left(1 + \gamma^2\left(\frac{2}{3}e^{1/\beta} + \frac{1}{3}e^{-2/\beta} - 1\right)\right), \tag{2}$$

*where $Q$ is the set of query nodes chosen from the public dataset, $\beta$ is the scale of the Laplace mechanism.*

*Proof Sketch.* First, using the Laplacian mechanism, the computation of the noisy label for each public query node is $1/\beta$-DP. We perform the transformation of the privacy variables using the RDP formula for the Laplacian mechanism. As the model uses only a random sample of the data to select the nodes for training the private model, we, therefore, apply the tight advanced composition of Theorem 8 to obtain $\varepsilon_{\text{LAP}o\text{POIS}}(\alpha)$. To obtain a bound corresponding to $Q$ queries, we further use the advanced composition theorem of RDP. The final expression is obtained by appropriate substitution of $\alpha$ and finally invoking Lemma 5 for the composed mechanisms. The detailed proof is provided in Appendix B.2. $\qquad\square$

**Remark.** We remark that Equation equation 2 provides only a rough upper bound. It is not suggested to use it for manual computation of $\varepsilon$ as it would give a much larger estimate. We provide it here for simplicity and to show the effect of the number of queries and sampling ratio on the final privacy guarantee. For our experiments, following Mironov (2017), we numerically compute the privacy budget with $\delta < 1/|V^\dagger|$ while varying $\alpha \in \{2, \ldots, 32\}$ and report the corresponding best $\varepsilon$.

## 4 Experimental Evaluation

With our experiments, we aim to answer the following research questions:

**RQ 1.** *How do the performance and privacy guarantees of PRIVGNN compare to that of the baselines?*

**RQ 2.** *What is the effect of sampling different K-nearest neighbors on the performance of private PRIVGNN method?*

**RQ 3.** *How does the final privacy budget of different approaches compare with an increase in the number of query nodes (|Q|)?*

**RQ 4.** *What is the effect of varying the sampling ratios on the privacy-utility trade-off for PRIVGNN?*

**RQ 5.** *How do various design choices of PRIVGNN affect its performance?*

Besides, we conduct additional experiments to investigate (i) the robustness of PRIVGNN towards two membership inference attacks (c.f. Section C), (ii) the sensitivity of PRIVGNN (c.f. Appendix D.2) to the size of private/public datasets and (iii) the effect of pre-training on the accuracy of PRIVGNN (c.f. Appendix D.7). We also discuss the time and space requirements of PRIVGNN in Appendix H and the behavior of PRIVGNN and PATE baselines when the features are more informative than the graph structure in Appendix D.1.

### 4.1 Experimental Setup

**Datasets.** We perform our experiments on four representative datasets, namely Amazon Shchur et al. (2018), Amazon2M Chiang et al. (2019), Reddit Hamilton et al. (2017) and Facebook Traud et al. (2012). We also perform additional experiments on ArXiv Hu et al. (2020) and Credit defaulter graph Agarwal et al. (2021) with highly informative features. The detailed description of the datasets is provided in Appendix F and their statistics in Table 11.

**Baselines.** We compare our PRIVGNN approach with three baselines. These are (i) *non-private inductive baseline (B1)* in which we train a GNN model on all private data and test the performance of the model on the public test set, (ii) *non-private transductive baseline (B2)* in which we train the GNN model using the public train split with the ground truth labels. The performance on the public test split estimates the "best possible" performance on the public data (iii) *two variants of PATE baseline* namely PATEM and PATEG, which use MLP and GNN respectively as their teacher models. A detailed description of the baselines is provided in Appendix G.

**Model and Hyperparameter Setup.** For the private and public GNN models, we employ a two-layer GraphSAGE model Hamilton et al. (2017) with a hidden dimension of 64 and ReLU activation function with batch normalization. For the MLP used in PATEM, we used three fully connected layers and ReLU activation layers. All methods are compared using the same hyperparameter settings described in Appendix G.1.

**Privacy Parameters.** We set the privacy parameter $\lambda = 1/\beta$. To demonstrate the effects of this parameter, we vary $\lambda$ and report the corresponding privacy budget for all queries. We set the reference values as follows $\{0.1, 0.2, 0.4, 0.8, 1\}$. We set $\delta$ to $10^{-4}$ for Amazon, Facebook, Credit, and Reddit, and $10^{-5}$ for ArXiv and Amazon2M. We fix the subsampling ratio $\gamma$ to 0.3 for all datasets except for Amazon2M, which is set to 0.1. All our experiments were conducted for 10 different instantiations, and we report the mean values across the runs in the main paper. The detailed results with the standard deviation are provided in Appendix A.

## 5  Results

We now discuss the results of our five research questions presented in Section 4.

### 5.1  Privacy-utility Trade-off (RQ 1)

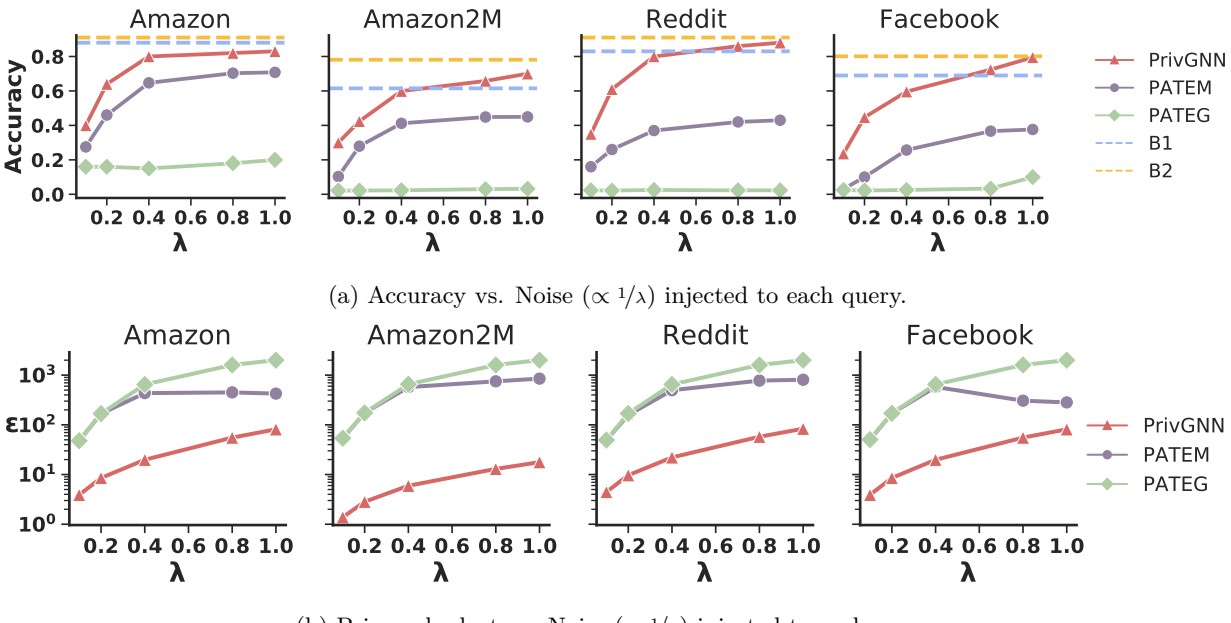

(a) Accuracy vs. Noise ($\propto 1/\lambda$) injected to each query.

(b) Privacy budget vs. Noise ($\propto 1/\lambda$) injected to each query.

Figure 2: Privacy-utility analysis. Here, $|Q|$ is set to 1000. For PRIVGNN, $\gamma$ is set to 0.1 for Amazon2M and 0.3 for other datasets.

Figure 2 shows the performance of our privacy-preserving PRIVGNN method as compared to two private and two non-private baselines. Since the achieved privacy guarantees depend on the injected noise level, we report the performance (c.f. Figure 2a) and the final incurred privacy budget (c.f. Figure 2b) with respect to $\lambda = 1/\beta$ which is inversely proportional to the injected noise for each query. Here, we set $|Q|= 1000$ for all three private methods.

First, the performance of our PRIVGNN approach converges quite fast to the non-private method, B1, as the noise level decreases. Note that B1 used all the private data for training and has no privacy guarantees. The reason for this convergence is that as the noise level decreases in PRIVGNN, the amount of noise introduced to the psuedo-labels decreases. As a result, the model can rely more on the less noisy psuedo-labels, leading to improved performance that approaches that of B1. The second non-private baseline, B2, trained using part of the public data, achieves slightly higher results than B1. This is because B2 benefits from the larger

size of the public dataset, which provides more diverse and representative samples for training. By having access to a larger and more varied set of data, B2 can capture a wider range of patterns and information, resulting in slightly improved performance compared to B1. For Amazon2M, Facebook and Reddit, PRIVGNN outperforms B1 for $\lambda > 0.4$.

Secondly, compared with the private methods (PATEM and PATEG), PRIVGNN achieves significantly better performance. For instance, at a smaller $\lambda = 0.2$, we achieved 40% and 300% improvement in accuracy (with respect to private baselines), respectively, on the Amazon dataset. We observe that for datasets with a high average degree (Amazon2M, Facebook, and Reddit), PRIVGNN achieved higher improvements of up to 134% and 2672% in accuracy when compared to the PATE methods. Moreover, as we will discuss next, PRIVGNN incurs significantly lower privacy costs as compared to the two private baselines.

Thirdly, we plot the incurred privacy budget $\varepsilon$ corresponding to different $\lambda$ in Figure 2b. PRIVGNN achieves a relatively small $\varepsilon$ of 2.83 on the Amazon2M dataset, while on the Amazon, Reddit, and Facebook dataset, a $\varepsilon$ of 8.53 at $\lambda = 0.2$. The corresponding value of $\varepsilon$ on PATEM and PATEG for the datasets are 167.82 on Amazon, 173.82 on Amazon2M, 157.90 for Reddit, and 170.41 for Facebook. *In fact, PRIVGNN achieves the performance of non-private baseline at $\varepsilon$ equal to 5.94 and 19.81 for Amazon2M and Reddit datasets respectively.*

It is not surprising that both PATEM and PATEG have similar $\varepsilon$ on all datasets except for Reddit. This is because PATE methods depend on agreement or consensus among the teacher models, which further reduces $\varepsilon$. This indicates that there is no consensus among the teachers. We observe that the privacy budget increases as the $\lambda$ gets larger. This phenomenon is expected since larger $\lambda$ implies low privacy and higher risk (high $\varepsilon$).

**Summary.** PRIVGNN outperforms both variants of PATE by incurring a lower privacy budget and achieving better accuracy. We attribute this observation to the following reasons. **First**, our privacy budget is primarily reduced due to the subsampling mechanism, whereas PATE uses the complete private data. **Second**, in the case of PRIVGNN, we train a personalized GNN for each query node using nodes closest to the query node and the induced relations among them. We believe that this leads to more accurate labels used to train the student model, resulting in better performance. On the other hand, the worse privacy and accuracy of PATE indicates a larger disagreement among the teachers. The random data partitioning in PATE destroys the graph structure. Moreover, teachers trained on disjoint and disconnected portions of the data might overfit specific data portions, hence the resulting disagreement. The Appendix (Table 5) contains the mean and standard deviation of the results shown in Figure 2a.

## 5.2 Effect of $K$ on the Accuracy (RQ 2)

To quantify the effect of the size of the random subset of private nodes on the accuracy of PRIVGNN, we vary the number of neighbors $K$ used in $K$-nearest neighbors. Note that the computed privacy guarantee is independent of $K$. For the Amazon dataset, we set $K$ to $\{300, 750\}$, and for the Amazon2M, Reddit, and Facebook dataset, we set $K$ to $\{750, 1000, 3000\}$. We select smaller $K$ for Amazon due to the small size of the private dataset.

As shown in Table 1, a general observation is that the higher the $K$, the higher the accuracy. For instance, sampling $K = 750$ nodes achieve over 50% improvements on the Amazon dataset than using $K = 300$ private nodes for training. On the Amazon2M, Reddit, and Facebook datasets, a value of $K$ above 1000 only increases the accuracy marginally. From the above results, we infer that the value of the hyperparameter $K$ should be chosen based on the average degree of the graph. A small $K$ suffices for sparse graphs (Amazon), while for graphs with a higher average degree (Amazon2M, Reddit, and Facebook), a larger $K$ leads to better performance. We further quantify the effect of $K$ in PRIVGNN by removing KNN and directly training with the entire private graph in our ablation studies(c.f. Section 5.5).

## 5.3 Effect of $|Q|$ on the Accuracy and Privacy (RQ 3)

Since the privacy budget is highly dependent on the number of queries answered by the teacher GNN model, we compare the performance and relative privacy budget incurred for answering different numbers of queries.

Table 1: Accuracy for varying $K$ and $|Q| = 1000$.

| | Amazon | | Amazon2M | | | Reddit | | | Facebook | | |
|---|---|---|---|---|---|---|---|---|---|---|---|
| $\lambda/K$ | 300 | 750 | 750 | 1000 | 3000 | 750 | 1000 | 3000 | 750 | 1000 | 3000 |
| 0.1 | 0.17 | 0.40 | 0.22 | 0.26 | 0.30 | 0.30 | 0.33 | 0.35 | 0.15 | 0.23 | 0.24 |
| 0.2 | 0.34 | 0.64 | 0.36 | 0.41 | 0.42 | 0.48 | 0.53 | 0.61 | 0.26 | 0.41 | 0.45 |
| 0.4 | 0.53 | 0.80 | 0.41 | 0.57 | 0.60 | 0.65 | 0.72 | 0.80 | 0.33 | 0.55 | 0.60 |
| 0.8 | 0.56 | 0.82 | 0.44 | 0.59 | 0.66 | 0.69 | 0.76 | 0.86 | 0.38 | 0.69 | 0.72 |
| 1.0 | 0.59 | 0.83 | 0.47 | 0.64 | 0.70 | 0.71 | 0.77 | 0.88 | 0.40 | 0.76 | 0.80 |

Figure 3: Accuracy for the different number of queries answered by the teacher model of PrivGnn

In Figure 3, we observe a negligible difference in accuracy when 1000 query nodes are pseudo-labeled and when only 500 queries are employed across all datasets at different noise levels except on Amazon2M. Further looking at detailed results in Table 2, for the different ranges of $\lambda$ on the Amazon, Reddit, and Facebook datasets, we observe up to 46% decrease in privacy budget of PrivGnn for answering 500 queries over answering 1000 queries. On the Amazon2M dataset, we observe up to 45% decrease in privacy budget. This implies that smaller $|Q|$ is desirable, which PrivGnn offers.

In Table 2, we also compare PrivGnn with PATE variants with respect to the incurred privacy budget $\varepsilon$ for answering the different number of queries. Our method offers a significantly better privacy guarantee (over 20 times reduction in $\varepsilon$) than PateM and PateG across all datasets. While decreasing the number of queries shows a negligible change in accuracy, it greatly reduces the incurred privacy budget for PrivGnn. In particular, the privacy budget (see Table 2), $\varepsilon$, of PrivGnn is reduced by 50% on the Amazon, Reddit, and Facebook datasets and by 40% on the Amazon2M dataset with only a slight reduction in accuracy. To determine the point at which the tradeoff between privacy and utility becomes severely compromised, we conducted additional experiments by adjusting the number of queries, using the additional reference values {300, 200, 100, 10}. The results of these experiments are available in Appendix D.6.

We remark that we also experimented with more intelligent strategies for query selection which did not show any performance gains. In particular, we employed structure-based approaches such as clustering and ranking based on centrality measures such as PageRank and degree centrality to choose the most representative query nodes but observed no performance gains over random selection.

## 5.4 Varying Subsampling Ratio $\gamma$ (RQ 4)

A smaller sampling ratio will reduce the amount of private data used for training which will, in turn, reduce the privacy budget $\varepsilon$. Therefore, to validate the effectiveness of the sampling ratio, we vary $\gamma$ from 0.1 to 0.3. This allows us to further benefit from the privacy amplification by subsampling as explained in Section B.1. Since $\gamma$ affects the amount of the available data used in training the teacher model, we only perform this experiment on representative datasets (Amazon2M and Reddit). Table 3 demonstrates that the accuracy achieved using a sampling ratio of $\gamma = 0.1$ on the Amazon2M dataset is only slightly lower than that of $\gamma = 0.3$ for $\lambda = 0.4$, with a decrease of only 6%. This indicates that the sampled data at $\gamma = 0.1$ captures

Table 2: Privacy budget ($\varepsilon$) (lower the better) for varying number of queries $|Q|$ answered by the teacher.

| | $|Q|$ | $\lambda \to$ | 0.1 | 0.2 | 0.4 | 0.8 | 1.0 |
|---|---|---|---|---|---|---|---|
| AMAZON | 500 | PRIVGNN | 2.67 | 5.69 | 13.15 | 31.10 | 44.07 |
| | | PATEM | 27.82 | 87.82 | 223.08 | 233.46 | 220.57 |
| | | PATEG | 27.82 | 87.82 | 327.82 | 800.97 | 1000.97 |
| | 1000 | PRIVGNN | 3.90 | 8.53 | 19.81 | 55.30 | 81.23 |
| | | PATEM | 47.82 | 167.82 | 434.69 | 449.36 | 424.78 |
| | | PATEG | 47.82 | 167.82 | 647.82 | 1600.97 | 1967.58 |
| AMAZON2M | 500 | PRIVGNN | 0.97 | 1.96 | 4.03 | 8.73 | 11.17 |
| | | PATEM | 33.82 | 93.82 | 294.89 | 379.92 | 416.65 |
| | | PATEG | 33.82 | 93.82 | 333.82 | 801.73 | 1000.73 |
| | 1000 | PRIVGNN | 1.39 | 2.83 | 5.94 | 12.98 | 17.73 |
| | | PATEM | 53.82 | 173.82 | 573.80 | 751.09 | 850.63 |
| | | PATEG | 53.82 | 173.82 | 653.82 | 1601.73 | 2001.73 |
| REDDIT | 500 | PRIVGNN | 2.67 | 5.69 | 13.15 | 31.10 | 44.07 |
| | | PATEM | 29.43 | 83.73 | 243.45 | 392.23 | 415.35 |
| | | PATEG | 29.43 | 89.43 | 329.43 | 801.17 | 1001.17 |
| | 1000 | PRIVGNN | 3.90 | 8.53 | 19.81 | 55.30 | 81.23 |
| | | PATEM | 49.43 | 157.90 | 496.61 | 775.95 | 808.43 |
| | | PATEG | 49.43 | 169.43 | 649.43 | 1601.17 | 2001.17 |
| FACEBOOK | 500 | PRIVGNN | 2.67 | 5.96 | 13.15 | 31.10 | 44.07 |
| | | PATEM | 30.41 | 90.41 | 293.44 | 151.02 | 136.41 |
| | | PATEG | 30.41 | 90.41 | 330.41 | 801.30 | 1000.30 |
| | 1000 | PRIVGNN | 3.90 | 8.53 | 19.81 | 55.30 | 81.23 |
| | | PATEM | 50.41 | 170.41 | 579.15 | 306.12 | 282.63 |
| | | PATEG | 50.41 | 170.41 | 650.41 | 1601.30 | 2001.30 |

similar underlying patterns and distributions to the data with $\gamma = 0.3$. On the Reddit dataset, we observe a 35% decrease in the accuracy when $\gamma = 0.1$ at the same $\lambda$. Nonetheless, we observe a 315% reduction in the privacy budget when $\gamma = 0.1$. This implies that smaller $\gamma$ offers better $\varepsilon$ on both datasets at a relatively lower drop in accuracy. We conclude that PRIVGNN provides an overall good privacy-utility trade-off with respect to $\gamma$. In other words, for larger datasets, a significant reduction in privacy costs can be achieved (when using smaller $\gamma$) with a relatively lower degradation in accuracy.

Table 3: Performance and the respective privacy budget for different sampling ratios $\gamma$ with $|Q| = 1000$ for Amazon2M and Reddit. The non-private baseline B1 achieves an accuracy of 0.62 and 0.78 for Amazon2M and Reddit, respectively. PRIVGNN already achieves comparable or better accuracy with $\varepsilon$ of 5.94 and 19.81, respectively (marked in bold).

| | $\gamma = 0.1$ | | | | $\gamma = 0.3$ | | | |
|---|---|---|---|---|---|---|---|---|
| | Reddit | | Amazon2M | | Reddit | | Amazon2M | |
| $\lambda$ | $\varepsilon$ | Accuracy | $\varepsilon$ | Accuracy | $\varepsilon$ | Accuracy | $\varepsilon$ | Accuracy |
| 0.1 | 1.20 | 0.19 | 1.39 | 0.30 | 3.90 | 0.35 | 4.45 | 0.35 |
| 0.2 | 2.47 | 0.37 | 2.83 | 0.42 | 8.53 | 0.61 | 9.69 | 0.51 |
| 0.4 | 5.20 | 0.52 | **5.94** | **0.60** | **19.81** | **0.80** | 22.11 | 0.64 |
| 0.8 | 11.82 | 0.63 | 12.98 | 0.66 | 55.30 | 0.86 | 57.60 | 0.69 |
| 1.0 | 15.44 | 0.65 | 17.17 | 0.70 | 81.23 | 0.88 | 83.53 | 0.77 |

## 5.5 Ablation Studies (RQ 5)

Table 4: Ablation studies. Here, we present the accuracy of different methods with varying $\lambda$ and $|Q| = 1000$. We show for representative datasets Amazon2M and Reddit with $\gamma = 0.1$ and 0.3 respectively.

| | $\lambda \rightarrow$ | 0.1 | 0.2 | 0.4 | 0.8 | 1.0 |
|---|---|---|---|---|---|---|
| Reddit | PrivGnn | 0.35 | 0.61 | 0.80 | 0.86 | 0.88 |
| | KNNRem | 0.34 | 0.58 | 0.75 | 0.79 | 0.81 |
| | SingleGNN | 0.28 | 0.54 | 0.72 | 0.77 | 0.79 |
| Amazon2M | PrivGnn | 0.30 | 0.42 | 0.60 | 0.66 | 0.70 |
| | KNNRem | 0.28 | 0.39 | 0.49 | 0.56 | 0.59 |
| | SingleGNN | 0.25 | 0.32 | 0.43 | 0.54 | 0.56 |

To answer RQ 5, we conducted ablation studies by varying different components of our proposed model. First, we removed the KNN component (Line 4 in Algorithm 1) and constructed the induced subgraph on the entire subsampled dataset for each query. We refer to this as KNNRem. Secondly, besides removing the KNN component, we only sample once from the private data instead of sampling for each query. Hence, we only train a single teacher model using the graph induced by subsampling. We refer to this as SingleGNN. We present our results in Table 4. We remark that, theoretically, the same privacy guarantees as our method holds for these two ablations. In practice, on the other hand, PrivGnn would offer better privacy because only a smaller portion of the private sample is used for teacher training compared to the ablations in which the KNN component is removed.

We make the following observations from our results. First, we see up to 18% and 8% drop in accuracy with KNNRem on the Amazon2M and Reddit datasets, respectively. This indicates the importance of our KNN approach since only the most informative node for the query node is used in training and predicting the pseudo-label.

Secondly, considering the performance of SingleGNN, the advantage of the multiple subsampling and the KNN components is prominent in both datasets. At a higher noise level (when $\lambda = 0.2$), we observe a performance drop of 23% and 12% on the Amazon2M and Reddit dataset, respectively, when SingleGNN is used as compared to PrivGnn.

**Additional experiments.** We performed several additional experiments. This includes two membership inference attacks to evaluate the privacy leakage of the released model. We observe that PrivGnn reduces the attacks to a random guess (Appendix C). We analyze the impact of informative node features on the performance of PrivGnn in Appendix D.1. The result indicates that PrivGnn in the presence of highly informative node features performs well than both PateM and PateG baselines. To better understand the influence of the private-public graph size on the performance of PrivGnn, we perform a sensitivity analysis by flipping the graphs. The result shows that PrivGnn is not sensitive to the size of the private-public graphs (Appendix D.2). We also perform an experiment that utilizes a pre-training technique to improve the performance of PrivGnn. The result shows a slight improvement in the performance of PrivGnn (Appendix D.7). In Appendix D.4, we compare our method to an additional non-private baseline. This baseline involves the joint training of a single GNN on both the private and public graphs. Furthermore, in Appendix D.5, we evaluate another private baseline where Laplacian noise is directly applied to the ground truth label of each query node. Finally, we analyze the runtime and space requirements of PrivGnn in Appendix H and highlight the limitations and future works (Appendix I).

## 6 Conclusion

We propose PrivGnn, a novel privacy-preserving framework for releasing GNN models with differential privacy guarantees. Our approach leverages the knowledge distillation framework, which transfers the knowledge of the *teacher* models trained on private data to the *student* model. Privacy is intuitively and

theoretically guaranteed due to private data subsampling as well as the noisy labeling of public data. Moreover, as we build *personalized* teacher models by using the $K$-nearest neighbors of the corresponding query nodes, the trained teacher model is more confident in its prediction. Results from the ablation studies further support our design choices. We present the privacy analysis of our approach by leveraging the Rènyi differential privacy framework. Our experiment results on six real-world datasets show the effectiveness of our approach in obtaining good accuracy under practical privacy guarantees.

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

## Appendix

**Organization.** The Appendix is organized as follows. We begin by providing the detailed accuracy scores of different methods in Appendix A. Missing theoretical details and proof are provided in Appendix B. We launch two black-box membership inference attacks in Appendix C followed by the analysis of the influence of highly informative features (Appendix D.1), the effect of the size of the public graph (Appendix D.2), a real-world example and motivation (Appendix D.3), additional baselines including the joint training on the private and public graph (Appendix D.4), adding noise directly to the groundtruth labels of the query nodes (Appendix D.5), privacy utility tradeoff with varying amount of queries (Appendix D.6), and the use of pre-training to improve performance (Appendix D.7). Appendix E outlines the distinctions between prior works and clarifies why our approach cannot be compared directly. Appendix F provides a detailed description of datasets used followed by details on baselines, hyperparameters, and training in Appendix G. The runtime and space requirement of PRIVGNN is discussed in Appendix H. Lastly, the limitations of the work are highlighted in Appendix I.

## A   Detailed Results

In Tables 5 and 6, we present the mean accuracy of the different methods as plotted in Figure 2 and 3 respectively.

Table 5: Performance of each method on all datasets. Note that the non-private baselines B1 and B2 are not changing with different $\lambda$.

| | Method | $\lambda = 0.1$ | $\lambda = 0.2$ | $\lambda = 0.4$ | $\lambda = 0.8$ | $\lambda = 1$ |
|---|---|---|---|---|---|---|
| Amazon | PRIVGNN | $0.40 \pm 0.08$ | $0.64 \pm 0.07$ | $0.80 \pm 0.13$ | $0.82 \pm 0.17$ | $0.83 \pm 0.10$ |
| | PATEM | $0.28 \pm 0.04$ | $0.46 \pm 0.03$ | $0.65 \pm 0.04$ | $0.70 \pm 0.03$ | $0.71 \pm 0.03$ |
| | PATEG | $0.16 \pm 0.05$ | $0.16 \pm 0.07$ | $0.15 \pm 0.08$ | $0.18 \pm 0.11$ | $0.20 \pm 0.09$ |
| | B1 | $0.88 \pm 0.02$ | $0.88 \pm 0.02$ | $0.88 \pm 0.02$ | $0.88 \pm 0.02$ | $0.88 \pm 0.02$ |
| | B2 | $0.91 \pm 0.02$ | $0.91 \pm 0.02$ | $0.91 \pm 0.02$ | $0.91 \pm 0.02$ | $0.91 \pm 0.02$ |
| Amazon2M | PRIVGNN | $0.30 \pm 0.05$ | $0.42 \pm 0.10$ | $0.60 \pm 0.18$ | $0.66 \pm 0.08$ | $0.70 \pm 0.19$ |
| | PATEM | $0.10 \pm 0.04$ | $0.28 \pm 0.02$ | $0.41 \pm 0.02$ | $0.44 \pm 0.01$ | $0.47 \pm 0.01$ |
| | PATEG | $0.02 \pm 0.01$ | $0.02 \pm 0.01$ | $0.03 \pm 0.03$ | $0.04 \pm 0.02$ | $0.04 \pm 0.03$ |
| | B1 | $0.62 \pm 0.01$ | $0.62 \pm 0.01$ | $0.62 \pm 0.01$ | $0.62 \pm 0.01$ | $0.62 \pm 0.01$ |
| | B2 | $0.78 \pm 0.01$ | $0.78 \pm 0.01$ | $0.78 \pm 0.01$ | $0.78 \pm 0.01$ | $0.78 \pm 0.01$ |
| Reddit | PRIVGNN | $0.35 \pm 0.09$ | $0.61 \pm 0.18$ | $0.80 \pm 0.10$ | $0.86 \pm 0.13$ | $0.88 \pm 0.07$ |
| | PATEM | $0.16 \pm 0.01$ | $0.26 \pm 0.01$ | $0.37 \pm 0.01$ | $0.42 \pm 0.03$ | $0.43 \pm 0.05$ |
| | PATEG | $0.03 \pm 0.03$ | $0.03 \pm 0.03$ | $0.06 \pm 0.02$ | $0.04 \pm 0.05$ | $0.08 \pm 0.04$ |
| | B1 | $0.83 \pm 0.01$ | $0.83 \pm 0.01$ | $0.83 \pm 0.01$ | $0.83 \pm 0.01$ | $0.83 \pm 0.01$ |
| | B2 | $0.92 \pm 0.01$ | $0.92 \pm 0.01$ | $0.92 \pm 0.01$ | $0.92 \pm 0.01$ | $0.92 \pm 0.01$ |
| Facebook | PRIVGNN | $0.24 \pm 0.08$ | $0.45 \pm 0.07$ | $0.60 \pm 0.15$ | $0.72 \pm 0.11$ | $0.80 \pm 0.11$ |
| | PATEM | $0.03 \pm 0.04$ | $0.10 \pm 0.05$ | $0.25 \pm 0.08$ | $0.37 \pm 0.03$ | $0.38 \pm 0.03$ |
| | PATEG | $0.02 \pm 0.03$ | $0.02 \pm 0.04$ | $0.03 \pm 0.04$ | $0.04 \pm 0.03$ | $0.10 \pm 0.03$ |
| | B1 | $0.69 \pm 0.01$ | $0.69 \pm 0.01$ | $0.69 \pm 0.01$ | $0.69 \pm 0.01$ | $0.69 \pm 0.01$ |
| | B2 | $0.80 \pm 0.01$ | $0.80 \pm 0.01$ | $0.80 \pm 0.01$ | $0.80 \pm 0.01$ | $0.80 \pm 0.01$ |

Table 6: Mean accuracy of varying the number of queries $|Q|$.

| | Amazon | | Amazon2M | | Reddit | | Facebook | |
|---|---|---|---|---|---|---|---|---|
| $\lambda/|Q|$ | 500 | 1000 | 500 | 1000 | 500 | 1000 | 500 | 1000 |
| 0.1 | 0.31 | 0.4 | 0.22 | 0.30 | 0.28 | 0.35 | 0.18 | 0.24 |
| 0.2 | 0.54 | 0.64 | 0.37 | 0.42 | 0.45 | 0.61 | 0.35 | 0.45 |
| 0.4 | 0.76 | 0.8 | 0.51 | 0.60 | 0.75 | 0.80 | 0.56 | 0.60 |
| 0.8 | 0.79 | 0.82 | 0.55 | 0.66 | 0.79 | 0.86 | 0.68 | 0.72 |
| 1.0 | 0.77 | 0.83 | 0.64 | 0.70 | 0.83 | 0.88 | 0.76 | 0.80 |

## B    Detailed Proof

Here, we provide missing theoretical details and the full proof of Theorem 7. In the following, we motivate the use of private data subsampling to achieve a reduction in the privacy budget. We use Theorem 8 in our analysis to bound the privacy budget (in the RDP framework) of our Poisson subsampled Laplacian mechanism.

### B.1    Privacy Amplification by Subsampling

A commonly used approach in privacy is *subsampling* in which the DP mechanism is applied to the randomly selected sample from the data. Subsampling offers a stronger privacy guarantee in that the one data point that differs between two neighboring datasets has a decreased probability of appearing in the smaller sample. That is, when we apply an $(\varepsilon, \delta)$-DP mechanism to a random $\gamma$-subset of the data, the entire procedure satisfies $(O(\gamma\varepsilon), \gamma\delta)$-DP. The intuitive notion of amplifying privacy by subsampling is that the privacy guarantee of the DP mechanism is tighter by applying it to a small random subsample of records from a given dataset. This is also referred to as the *subsampling lemma* in the literature Balle et al. (2018). Under some restrictions on $\alpha$, we can represent the combination of the subsampling lemma, and the tight advanced composition of RDP Wang et al. (2019); Zhu & Wang (2019) as: $\varepsilon_{\mathcal{M}oSample_\gamma}(\alpha) \leq O(\gamma^2 \varepsilon_{\mathcal{M}}(\alpha))$. In this paper, we apply the Poisson subsampled RDP-amplification bound from Zhu & Wang (2019).

**Theorem 8 (General upper bound Zhu & Wang (2019)).** *Let $\mathcal{M}$ be any mechanism that obeys $(\alpha, \varepsilon(\alpha))$-RDP. Let $\gamma$ be the subsampling probability, then for integer $\alpha \geq 2$, the privacy budget is*

$$\varepsilon_{MoPoissonSample}(\alpha) \leq \frac{1}{\alpha - 1} \log \left\{ (1 - \gamma)^{\alpha - 1}(\alpha\gamma - \gamma + 1) + \binom{\alpha}{2} \gamma^2 (1 - \gamma)^{\alpha - 2} e^{\varepsilon(2)} \right.$$
$$\left. + 3 \sum_{\ell = 3}^{\alpha} \binom{\alpha}{\ell} (1 - \gamma)^{\alpha - \ell} \gamma^\ell e^{(\ell - 1)\varepsilon(\ell)} \right\}.$$

### B.2    Proof of our Privacy Guarantee

*Proof of Theorem 7.* Our algorithm is composed of (i) a sampling mechanism in which a small sample of private data is used to train the private GNN model and (ii) a Laplacian mechanism (with scale parameter $\beta$), which is used to generate a noisy label for the public node (queried on corresponding private GNN).

First, note that as the $L_1$ norm of the posterior is bounded by 1, we add independent Laplacian noise to each posterior element with scale $\beta$ (Note that each posterior element is also bounded by 1). The computation of noisy label for each public query node is, therefore, $1/\beta$-DP. The sequential composition Dwork et al. (2006) over $N$ queries will result in a crude bound over the DP guarantee of our approach, namely $N/\beta$. To obtain a tighter bound that takes the effect of the private data subsampling into account, we perform the transformation of the privacy variables using the RDP formula for the Laplacian mechanism for $\alpha > 1$

$$\varepsilon_{\text{LAP}}(\alpha) = \frac{1}{\alpha - 1} \log \left( \left( \frac{\alpha}{2\alpha - 1} \right) e^{\frac{\alpha - 1}{\beta}} + \left( \frac{\alpha - 1}{2\alpha - 1} \right) e^{\frac{-\alpha}{\beta}} \right). \tag{3}$$

Moreover, the model uses only a random sample of the data to select the nodes for training the private model. We, therefore, apply the tight advanced composition of Theorem 8 to obtain an upper bound of $\varepsilon_{\text{LAP}o\text{POIS}}(\alpha)$. To simplify the expression, we set $\alpha = 2$ in the corresponding bound and obtain

$$\varepsilon_{\text{LAP}o\text{POIS}}(2) \leq \log \left( 1 - \gamma^2 + \gamma^2 e^{\varepsilon_{\text{LAP}}(2)} \right). \tag{4}$$

Substituting $\varepsilon_{\text{LAP}}(2)$ in equation 4, we obtain

$$\varepsilon_{\text{LAP}o\text{POIS}}(2) = \log \left( 1 - \gamma^2 + \gamma^2 \left( \frac{2}{3} \cdot e^{1/\beta} + \frac{1}{3} \cdot e^{-2/\beta} \right) \right). \tag{5}$$

Now applying the advanced composition for RDP (Lemma 6) for $|Q|$ queries, we get the upper bound on the total privacy loss of our approach at $\alpha = 2$

$$\varepsilon_{\text{PRIVGNN}}(2) \leq |Q|\log\left(1 - \gamma^2 + \gamma^2\left(\frac{2}{3} \cdot e^{1/\beta} + \frac{1}{3} \cdot e^{-2/\beta}\right)\right).$$

For any given $\delta > 0$, we use Lemma 5 and Eq. equation 1 to obtain the $(\varepsilon, \delta)$-DP guarantee. Specifically for any $\delta > 0$ we obtain

$$\varepsilon(\delta) = \min_{\alpha > 1} \frac{\log(1/\delta)}{\alpha - 1} + \varepsilon_{\text{PRIVGNN}}(\alpha - 1). \tag{6}$$

Substituting $\alpha = 3$ in the above we obtain the stated upper bound, i.e.,

$$\varepsilon(\delta) \leq \log\left(\frac{1}{\sqrt{\delta}}\right) + |Q|\log\left(1 + \gamma^2\left(\frac{2}{3}e^{1/\beta} + \frac{1}{3}e^{-2/\beta} - 1\right)\right),$$

thereby completing the proof. $\qquad\square$

## C  PrivGNN and Membership Inference Attacks

In addition to the theoretical guarantee of differential privacy, here, we empirically test the robustness of PRIVGNN against membership inference (MI) attacks. In particular, we apply the attack of Olatunji et al. (2021) in two different settings. In the first setting (Attack-1), the goal of the adversary is to distinguish private nodes from public nodes. In contrast, in the second setting (Attack-2), the goal of the adversary is to distinguish the public nodes which were privately labeled from the unlabeled public nodes.

Overall our two MI attacks consist of three phases: the training of a shadow model, the training of an attack model, and the membership inference phase, which involves using the trained attack model to infer the membership status of any given query from the target (attacked) model. In summary, we first generate posteriors for the shadow dataset using the shadow model. This generated data is then used to train the attack model, which is a binary classifier that distinguishes between the training and non-training nodes. For more details about the MI attack, we refer the reader to Olatunji et al. (2021) and their publicly available implementation.

In the following, we describe our two attack variants and the corresponding results: Attack-1 aims at determining the private nodes (the original training data), and Attack-2 aims to determine the pseudo-labeled public nodes (the noisy training set for the public student model). For all the attacks, we select the student model with the largest $\lambda = 1$, which corresponds to a lower privacy guarantee and the highest information leakage based on the achieved $\epsilon$ as the target model.

### C.1  Identifying Private Nodes via MI (Attack-1)

Here, we assume that the attacker does not know the training paradigm of the released student model but is aware of the architecture of the released model (target model). She wants to infer the private nodes used for training from the dataset available to her (for example, by some means, she was able to get hold of the Facebook social network). After training her attack model, the attacker queries the target model with nodes of interest and inputs her posteriors into her trained attack model to infer membership.

We compare the performance of the attack on a non-private model (B1) and PRIVGNN. As shown in Figure 4a, in the non-private model, the attacker can accurately identify private nodes which were used in training the target GNN model with high AUROC ($> 0.75$) on all datasets. However, the performance of that attack on PRIVGNN is reduced to a random guess ($\leq 0.54$).

### C.2  Identifying Private Nodes by Proxy via MI (Attack-2)

For this attack, we assume that the attacker knows that the released student model (target model) is trained using the knowledge distillation paradigm. Therefore, for such a strong attacker, her goal is to determine the

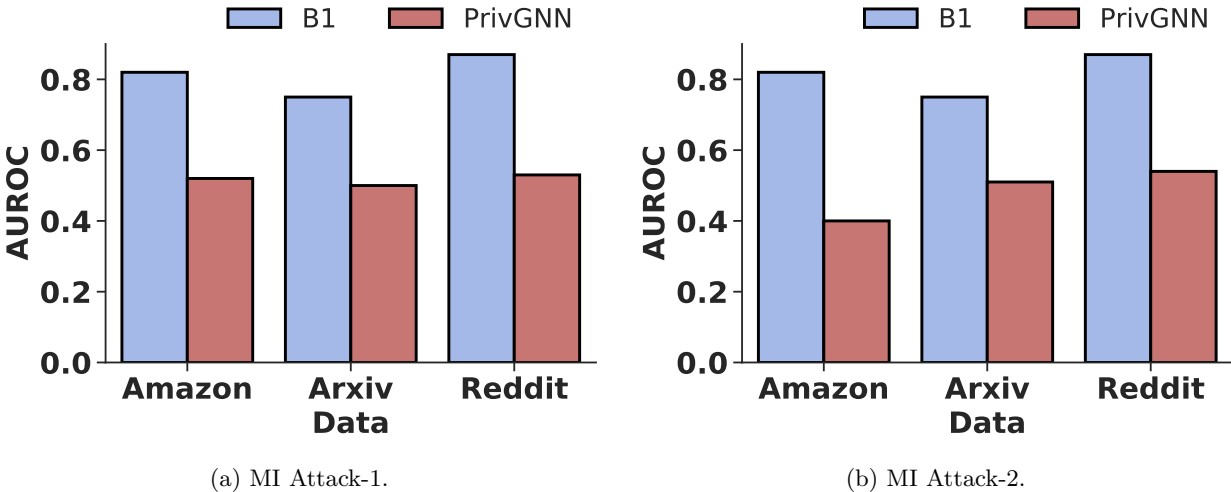

(a) MI Attack-1.                    (b) MI Attack-2.

Figure 4: Performance of MI Attack-1 and Attack-2 on PRIVGNN. B1 refers to the MI attack on the non-private GNN model.

public nodes that were privately labeled and used by PRIVGNN in training the target model from the other public nodes. The rationale is that if an attacker can confidently determine the public nodes that are privately (noisily) labeled, then it might reveal some information about the private dataset. For instance, consider an attacker that is aware that PRIVGNN utilizes *KNN* in selecting the private nodes used in labeling a public node. If she determines such a public node that was privately labeled, she might infer private information of up to $K$ private nodes assuming that she has access to the entire graph but does not know which is private or public. Therefore, the private nodes are said to be identified by "proxy".

As shown in Figure 4b, our PRIVGNN method reduces the MI attack to a random guess achieving an AUROC of 0.40, 0.51, and 0.54 on the Amazon, ArXiv, and Reddit datasets, respectively, as compared to the non-private MI attack (corresponding to the MI attack on Baseline B1) with AUROC of 0.82, 0.75, and 0.87. This is because of the two noise mechanisms that PRIVGNN adopts. Specifically, the random subsampling limits the private data used for the KNN procedure. Moreover, the knowledge transferred to the student model is limited due to noisy pseudo-labels.

## D  Additional Experiments

### D.1  Impact of Highly Informative Node Features

Here, we use two datasets, ArxiV and Credit, whose features are already highly informative for the task. To verify this, we first train the datasets on a three-layer MLP using only the features, and we obtained relatively high accuracy as compared to B2, which in addition uses the graph structure, as shown in Table 7.

As shown in Figure 5, on both datasets, the performance of PATEM is on par with PRIVGNN. On the ArXiv dataset, we achieved a 15% decrease over PATEM (when $\lambda = 0.2$). Note that the teacher models of PATEM are MLPs and do not utilize graph structure. The observed result is probably because the average node degree of ArXiv is very small ($\leq 2$). Thus, GNNs, which use the aggregation of the neighbors of a node to make predictions, might not benefit from such low-degree graphs. However, against PATEG, we achieved a 1169% improvement in performance which uses graph structure. Recall that PATEG teacher

Table 7: MLP accuracy of Credit and ArXiv. B2 utilizes the GNN model

|      | Credit | ArXiv |
|------|--------|-------|
| **MLP** | 0.78 | 0.55 |
| **B2**  | 0.81 | 0.63 |

models are queried on graph data when they are trained like MLP (due to the low connectivity among training nodes), which explains its worse performance than PATEM. Moreover, on the Credit dataset, the

privacy budget is very low, indicating high agreement among the teacher models in PATEM. We also observe that PATEG performs better on this dataset which further supports our argument that the feature is highly informative as compared to the performance on other datasets in Figure 2. We will like to emphasize that the drop in the privacy budget of the Credit dataset as less noise is added is not surprising. This is due to the data-dependent analysis that PATEM utilizes (see Papernot et al. (2016)).

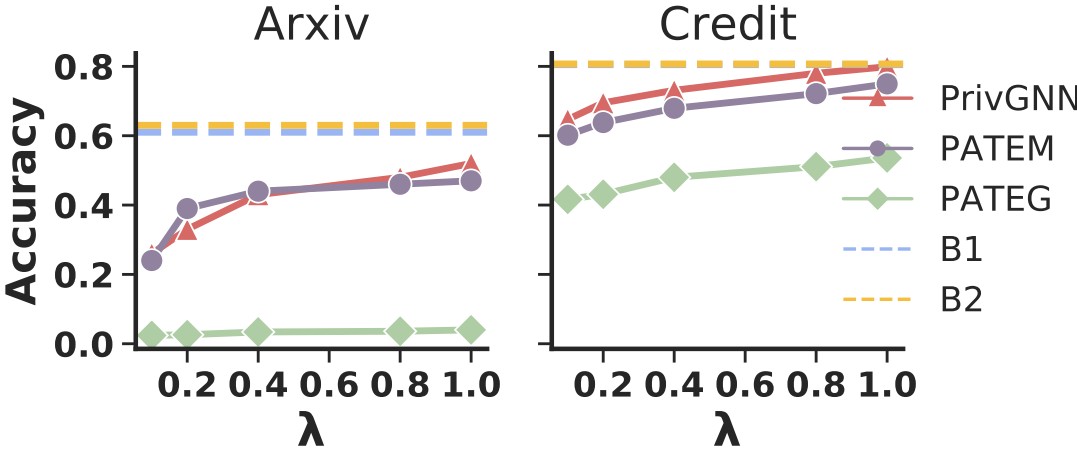

(a) Accuracy vs. Noise ($\propto 1/\lambda$) injected to each query.

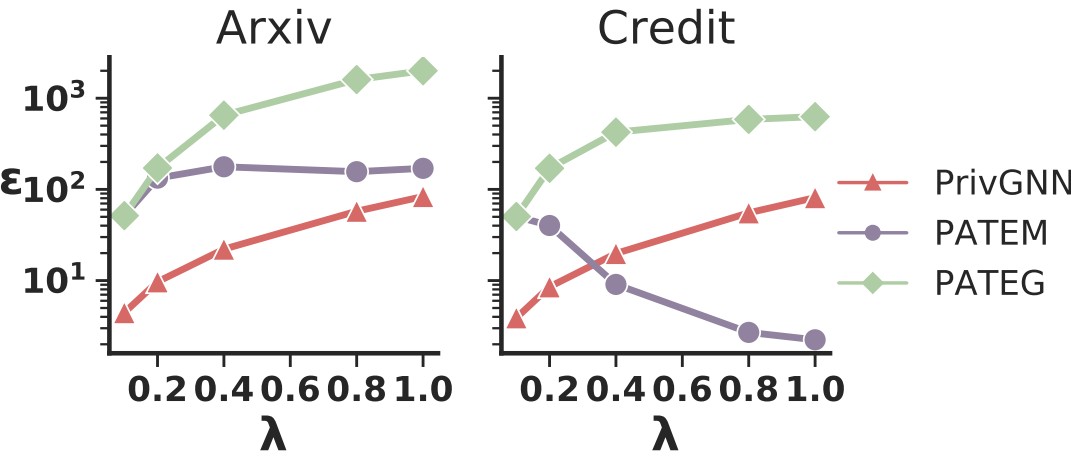

(b) Privacy budget vs. Noise ($\propto 1/\lambda$) injected to each query.

Figure 5: Privacy-utility analysis of the impact of highly informative node features. Here, $|Q|$ is set to 1000. For PRIVGNN, $\gamma$ is set to 0.3.

### D.2 Effect of the Size of the Public Dataset

To investigate the effect of the size of the public dataset on the performance of PRIVGNN, we flip the public and private portions corresponding to different datasets. For instance, instead of having 2500 nodes in the private dataset as shown for Amazon in Table 11, we have 6000 nodes for the private dataset and 2500 for the public dataset. As shown in Figure 6, there are no significant differences in the results of our model when there is a larger public dataset available (c.f Figure 2a). For a fair comparison, we run all baselines, including PATEM and PATEG, on the flipped data. PRIVGNN outperforms all private baselines on the flipped datasets. As expected, the result of the non-private baselines (B1 and B2) are interchanged. We observe that our model is not sensitive to the size of the private or public datasets. Therefore, our method is suitable in settings where the available private or public data is large and in settings with a limited private or public dataset.

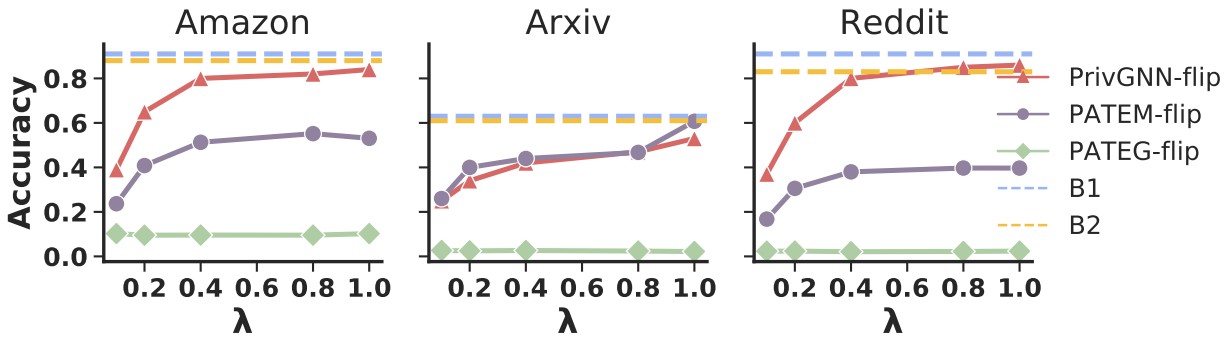

Figure 6: Performance of different models after flipping the public and private datasets.

### D.3    Real-world Scenario / Motivation

Our framework has a range of applications in real-world scenarios where public and private graphs exist simultaneously. One example of such a scenario is the Wikidata (Vrandečić & Krötzsch, 2014), which is a publicly available knowledge graph that is constructed using Wikipedia's content and can be easily downloaded. In contrast, the Google knowledge graph (Singhal, 2012) (which is a superset of Wikidata) is a private knowledge graph that is not openly accessible to the public. The implementation details of the Google knowledge graph are not officially documented, and the raw data is not available to anyone Ehrlinger & Wöß (2016). The Google knowledge graph can only be accessed through an API, which provides black-box access. However, it's worth noting that the Google knowledge graph is partly based on the contents of Freebase, which was later acquired by Wikidata Pellissier Tanon et al. (2016). Therefore, both public and private graph data exist in real-world scenarios and can potentially be utilized by the owner of private data (in this case Google) to train and deploy a private GNN model.

Similarly, in finance, there may be sensitive financial information that needs to be kept private, but combining this information with public economic data can lead to valuable insights for investors and policymakers.

### D.3.1    Application Example

One motivation for our approach is the practical content labeling problem Morrow et al. (2022); Saltz et al. (2021); Gao et al. (2018). Content labeling is a form of content moderation where labels are assigned to each content. Content labeling helps deal with misinformation, conspiracy theories, and misleading content (disinformation) that may affect day-to-day activities like voting, health behaviors, and promoting hate speech.

As an example, social network companies like Facebook have millions of users(nodes) and connections(edges). The profiles of some users are public and can be manually curated. For a specific snapshot (a point in time), Facebook assigns attributes (e.g., labels) to a subset of the users based on their contents or activities on the social network to identify bots and real users to tackle disinformation. Furthermore, Facebook could release a content labeling model trained on their private data (e.g., as an API) without endangering the participants' privacy or releasing trade secrets.

If a model trained on private data, such as a content labeling model, is released without any privacy protection, it could potentially violate intellectual property or trade secrets as manually annotating/assigning attributes to data can be expensive. Moreover, such a release could also put the privacy of participants at risk by enabling attacks like membership inference attacks. Therefore, it is crucial to have a framework that combines both public and private information while accounting for the classic adversarial game where the adversary tries to recover private data from the model that is released publicly using data from a similar distribution, like a different snapshot of Facebook.

With tech companies competing to gain a competitive advantage through the use of advanced machine learning and artificial intelligence techniques, there is an arms race to launch AI models trained on their

collected data. However, this has also raised concerns about the privacy of the individuals whose data is being used, as well as the potential for misuse of the models by those who have access to them. Our approach has great potential to be adopted by such companies looking to balance the need for advanced machine learning techniques with the need to protect the privacy of their users' data.

### D.4 Joint Training of the Private and Public Graph

As an additional non-private baseline, we conduct a joint training on both the private and public graphs, which we refer to as baseline **B3**. A single GNN is trained on with all nodes in the private graph and the train nodes of the public graph. Note that all the edges of both the private and public graph are used for training. The model is evaluated on the test set of the public graph, which is the same test set used for all experiments and baselines.

The results of the joint training (B3) and B2 are comparable, as shown in Table 8, with some cases being the same. This joint training supports our hypothesis that the B2 accuracy is the "best possible" estimate of the graph performance (see Baselines in Section 4.1).

Table 8: Performance of all baselines including the joint training baseline (B3).

| Data | **B1** | **B2** | **B3** |
|---|---|---|---|
| Amazon | $0.88 \pm 0.02$ | $0.91 \pm 0.02$ | $0.93 \pm 0.02$ |
| Amazon2M | $0.62 \pm 0.01$ | $0.78 \pm 0.01$ | $0.78 \pm 0.02$ |
| ArXiv | $0.61 \pm 0.01$ | $0.63 \pm 0.01$ | $0.65 \pm 0.01$ |
| Reddit | $0.83 \pm 0.01$ | $0.92 \pm 0.01$ | $0.92 \pm 0.02$ |
| Facebook | $0.69 \pm 0.01$ | $0.80 \pm 0.01$ | $0.83 \pm 0.02$ |
| Credit | $0.81 \pm 0.01$ | $0.81 \pm 0.01$ | $0.81 \pm 0.01$ |

### D.5 Adding Noise Directly to the Groundtruth Label of Query Nodes

To further evaluate the performance of PRIVGNN against a noisy "oracle", we add Laplace noise directly to the groundtruth label of each query nodes. We call this baseline **B4**. We note that this baseline is not feasible in our setting since the groundtruth label of the public model is not obtainable. Hence, a noisy oracle. The procedure is as follows. We first represent the true groundtruth label of the query nodes as a one-hot encoded vector of all zeros except in the position of the groundtruth label. We then add Laplacian noise to the one-hot encoded vector. We set the negative values of the noisy label vector to zero and normalize the obtained vector. The normalized label vector for each query is then used in training the student model. We show the privacy-utility tradeoff of PRIVGNN and B4 in Table 9. The accuracy of B4 is not surprising since noise is directly added to the true groundtruth labels. On all datasets, the performance of B4 is similar or slightly better than PRIVGNN. However, the extremely high privacy budget of B4 renders it useless. Such crude approach, although unrealistic, can achieve higher performance but also at a higher value of $\epsilon$.

### D.6 Privacy-utility Tradeoff by Varying the Number of Queries

To observe the privacy-utility tradeoff based on the number of queries answered by the teacher model, we fix the value of noise parameter $\lambda = 0.4$ and vary number of queries in the range $\{1000, 500, 300, 200, 100, 10\}$. Note that this experiment is an extended version of Section 5.3 in which we vary the number of answered queries from 500 to 1000. To visually observe the privacy-utility tradeoff at for different number of queries, we plot the accuracy on the left y-axis and the corresponding privacy in Figure 7. On all datasets, we observe that lower queries leads to lower accuracy and higher privacy guarantee (low epsilon). Furthermore, we observe a severe degradation in performance when the number of queries is below 400. Although the privacy guarantee is appealing but unsurprisingly the utility suffers because of noisy and limited training data used to train the public data.

### D.7 Improving Performance via Pre-training

As shown in Section D.2, PRIVGNN is not sensitive to the size of the public or private data. However, in settings where a larger public dataset is available, we ask whether the student model can benefit from such large "unlabeled" data. Therefore, we propose to leverage the abundance of the unlabeled dataset by

Table 9: Privacy-utility tradeoff of PRIVGNN and B4 (adding noise directly to the groundtruth output). Note that B4 does not change with $\lambda$.

| | | $\lambda \rightarrow$ | 0.1 | 0.2 | 0.4 | 0.8 | 1.0 |
|---|---|---|---|---|---|---|---|
| AMAZON | ACCURACY | PRIVGNN | $0.40 \pm 0.08$ | $0.64 \pm 0.07$ | $0.80 \pm 0.13$ | $0.82 \pm 0.17$ | $0.83 \pm 0.10$ |
| | | B4 | $0.42 \pm 0.05$ | $0.65 \pm 0.04$ | $0.82 \pm 0.03$ | $0.84 \pm 0.05$ | $0.85 \pm 0.07$ |
| | $\epsilon$ | PRIVGNN | 3.90 | 8.53 | 19.81 | 55.3 | 81.23 |
| | | B4 | 100 | 200 | 300 | 400 | 500 |
| AMA-ZON2M | ACCURACY | PRIVGNN | $0.30 \pm 0.05$ | $0.42 \pm 0.10$ | $0.60 \pm 0.18$ | $0.66 \pm 0.08$ | $0.70 \pm 0.19$ |
| | | B4 | $0.32 \pm 0.04$ | $0.46 \pm 0.05$ | $0.63 \pm 0.07$ | $0.68 \pm 0.04$ | $0.77 \pm 0.02$ |
| | $\epsilon$ | PRIVGNN | 1.38 | 2.82 | 5.94 | 12.97 | 17.73 |
| | | B4 | 100 | 200 | 300 | 400 | 500 |
| ARXIV | ACCURACY | PRIVGNN | $0.26 \pm 0.03$ | $0.33 \pm 0.02$ | $0.43 \pm 0.05$ | $0.48 \pm 0.04$ | $0.52 \pm 0.05$ |
| | | B4 | $0.28 \pm 0.02$ | $0.38 \pm 0.01$ | $0.46 \pm 0.03$ | $0.51 \pm 0.02$ | $0.54 \pm 0.04$ |
| | $\epsilon$ | PRIVGNN | 4.45 | 9.69 | 22.11 | 57.60 | 83.53 |
| | | B4 | 100 | 200 | 300 | 400 | 500 |
| REDDIT | ACCURACY | PRIVGNN | $0.35 \pm 0.09$ | $0.61 \pm 0.18$ | $0.80 \pm 0.10$ | $0.86 \pm 0.13$ | $0.88 \pm 0.07$ |
| | | B4 | $0.35 \pm 0.04$ | $0.63 \pm 0.03$ | $0.82 \pm 0.04$ | $0.87 \pm 0.02$ | $0.88 \pm 0.03$ |
| | $\epsilon$ | PRIVGNN | 3.90 | 8.53 | 19.81 | 55.3 | 81.23 |
| | | B4 | 100 | 200 | 300 | 400 | 500 |
| FACE-BOOK | ACCURACY | PRIVGNN | $0.24 \pm 0.08$ | $0.45 \pm 0.07$ | $0.60 \pm 0.15$ | $0.72 \pm 0.11$ | $0.80 \pm 0.11$ |
| | | B4 | $0.28 \pm 0.06$ | $0.47 \pm 0.03$ | $0.63 \pm 0.08$ | $0.74 \pm 0.05$ | $0.83 \pm 0.07$ |
| | $\epsilon$ | PRIVGNN | 3.90 | 8.53 | 19.81 | 55.3 | 81.23 |
| | | B4 | 100 | 200 | 300 | 400 | 500 |
| CREDIT | ACCURACY | PRIVGNN | $0.65 \pm 0.02$ | $0.70 \pm 0.05$ | $0.73 \pm 0.05$ | $0.78 \pm 0.04$ | $0.80 \pm 0.06$ |
| | | B4 | $0.65 \pm 0.01$ | $0.71 \pm 0.02$ | $0.73 \pm 0.04$ | $0.79 \pm 0.02$ | $0.80 \pm 0.02$ |
| | $\epsilon$ | PRIVGNN | 3.90 | 8.53 | 19.81 | 55.3 | 81.23 |
| | | B4 | 100 | 200 | 300 | 400 | 500 |

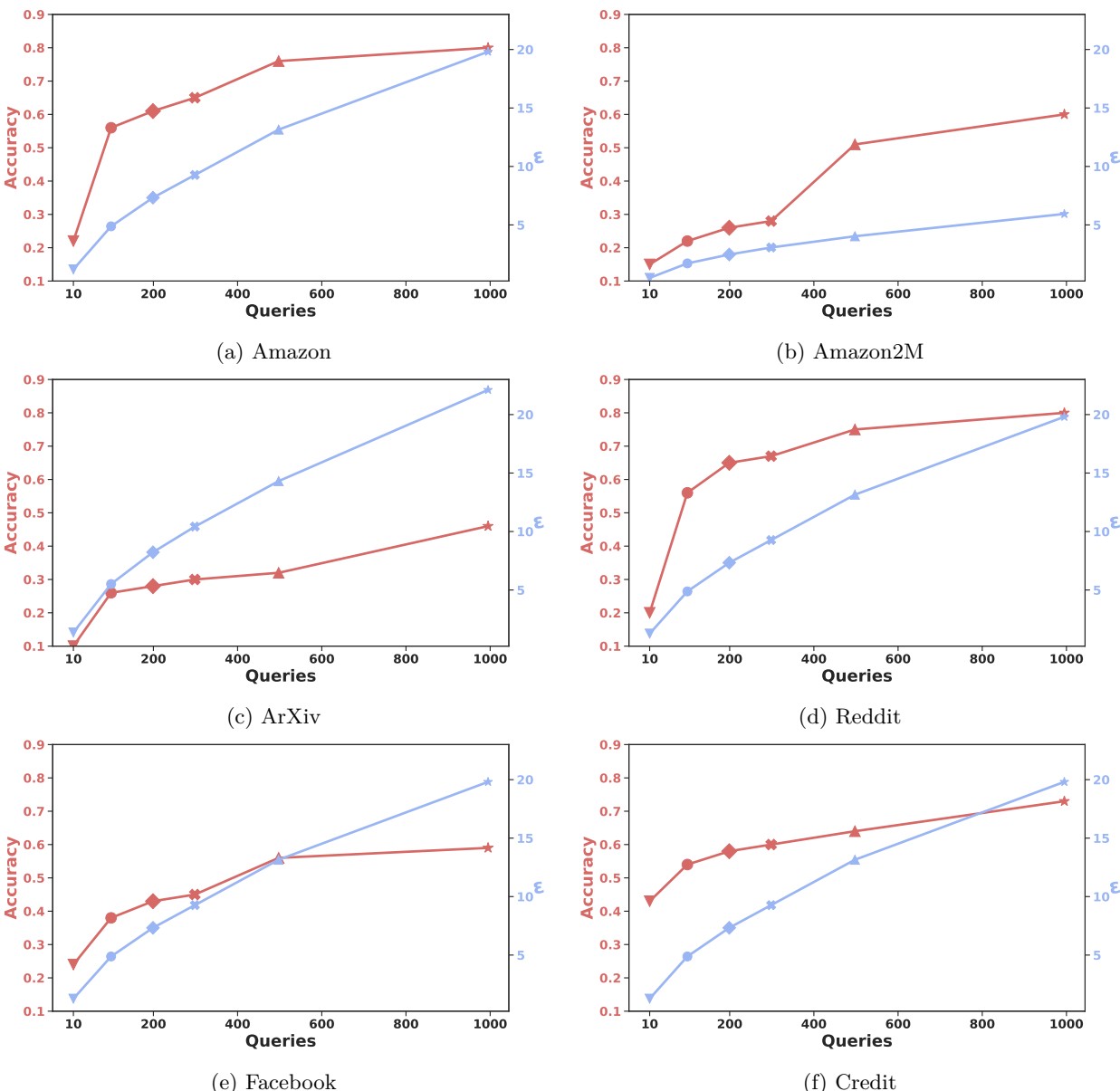

Figure 7: Privacy-utility tradeoff for different values of $|Q|$. We fix $\lambda = 0.4$ and $\gamma = 0.3$ for all datasets except for Amazon2M with $\gamma = 0.1$. The accuracy is represented by the red line on the left y-axis, while the achieved privacy is indicated by the blue line on the right y-axis. The markers on both lines correspond to the privacy-utility tradeoff for a specific number of queries.

pre-training the student GNN model using unsupervised objectives on the public dataset. The unsupervised objective is based on the graph reconstruction objective Hamilton et al. (2017) which assigns similar embedding to connected nodes. After pre-training, we initialize the student model with the pre-trained model and fine-tune the student model using the privately labeled nodes. We emphasize that the pre-training does not constitute any privacy risks since the data is public. One major advantage of unsupervised pre-training is that it acts as a regularizer and provides better generalization than randomly initializing the weights Erhan et al. (2010). We conducted a preliminary experiment on the Amazon2M and ArXiv datasets. As shown in Table 10, on a smaller number of queries ($|Q|$=300), we obtain an increment of up to 35% on Amazon2M and 17% on ArXiv over the randomly initialized PRIVGNN. We also observe up to 4% and 9% increase on higher number of queries ($|Q|$=1000) on Amazon2M and ArXiv, respectively.

As pointed out by Hu et al. (2019), pre-training GNNs is still relatively difficult and may require domain expertise to carefully select examples that are well correlated to the downstream task to avoid "negative" transfer (hurt generalization). Therefore, we leave an in-depth study of how pre-training can help the better performance of PRIVGNN on more datasets as future work.

Table 10: Performance of pre-training the student model on Amazon2M and ArXiv dataset. $\Delta$ indicates the % difference between PRIVGNN without pre-training and the pre-trained PRIVGNN (Pre-PRIVGNN).

|  | $|Q|$ | $\lambda \rightarrow$ | 0.1 | 0.2 | 0.4 | 0.8 | 1.0 |
|---|---|---|---|---|---|---|---|
| AMAZON2M | 1000 | PRIVGNN | 0.30 | 0.42 | 0.60 | 0.66 | 0.70 |
|  |  | PRE-PRIVGNN | 0.31 | 0.44 | 0.61 | 0.68 | 0.72 |
|  |  | $\Delta$ | 3% | 4% | 2% | 3% | 3% |
|  | 500 | PRIVGNN | 0.22 | 0.37 | 0.51 | 0.55 | 0.64 |
|  |  | PRE-PRIVGNN | 0.24 | 0.40 | 0.53 | 0.58 | 0.67 |
|  |  | $\Delta$ | 9% | 8% | 4% | 5% | 5% |
|  | 300 | PRIVGNN | 0.12 | 0.19 | 0.28 | 0.34 | 0.36 |
|  |  | PRE-PRIVGNN | 0.17 | 0.22 | 0.30 | 0.36 | 0.43 |
|  |  | $\Delta$ | 35% | 15% | 7% | 6% | 18% |
| ARXIV | 1000 | PRIVGNN | 0.26 | 0.33 | 0.43 | 0.48 | 0.52 |
|  |  | PRE-PRIVGNN | 0.28 | 0.36 | 0.47 | 0.51 | 0.54 |
|  |  | $\Delta$ | 7% | 9% | 9% | 6% | 4% |
|  | 500 | PRIVGNN | 0.21 | 0.28 | 0.32 | 0.39 | 0.41 |
|  |  | PRE-PRIVGNN | 0.22 | 0.30 | 0.34 | 0.42 | 0.42 |
|  |  | $\Delta$ | 4% | 7% | 6% | 7% | 2% |
|  | 300 | PRIVGNN | 0.10 | 0.16 | 0.30 | 0.34 | 0.36 |
|  |  | PRE-PRIVGNN | 0.12 | 0.19 | 0.34 | 0.37 | 0.40 |
|  |  | $\Delta$ | 18% | 17% | 12% | 8% | 11% |

## E    Differences with Existing Works

The following section explicitly outlines the differences between our approach and existing works.

### E.1    Differences with LPGNN Sajadmanesh & Gatica-Perez (2020)

In LPGNN, noise is directly added to both the node features and the true labels via local DP. Therefore, their derived DP guarantee is based on the multi-bit mechanism that adds noise to the node features and randomized response mechanism that adds noise to the true labels. This makes LPGNN not directly comparable with the privacy guarantee of PRIVGNN. LPGNN primarily aims to achieve local differential privacy for individual node features whereas PRIVGNN focuses on the global differential privacy for the entire graph and the released GNN model.

### E.2    Differences with VFGNN Zhou et al. (2020)

Although VFGNN assumes that the computational graph of GNN is split into two where the private data-related computations are handled by the data holders, and the non-private data-related computations is handled by a semi-honest server, their approach is largely disparate from PRIVGNN.

First, the initial embedding of the nodes of each data holder is computed by applying a secure multi-party computation (SMC) technique individually. Then, a local embedding is generated by aggregating the features of neighboring nodes, and either James-Stein Estimator or Gaussian noise is added directly to the local embedding to ensure DP. Finally, the local embedding is later combined for different data holders to form a global node embedding.

One possible modification to compare the performance of VFGNN with PRIVGNN is to remove the SMC technique, which is the key strength of their privacy framework, and obtain the local embedding without this cryptographic measure. However, this modification has a major drawback as it requires adding Gaussian noise directly to the local embedding to guarantee local differential privacy, which is not as strong as the privacy guarantee of PRIVGNN. Therefore, it is not feasible to make a fair comparison between the two methods.

### E.3 Differences with FedGNN Wu et al. (2021)

The FedGNN framework relies on each data holder possessing a complete dataset consisting of nodes, edges, and labels. The first step in FedGNN involves training local GNNs with the original data and correct labels for each data holder to obtain local gradients. To ensure differential privacy, noise is added to the local gradients. FedGNN adopts a horizontal federated learning approach, where parties collaborate on the same task while retaining their data locally.

On the other hand, PRIVGNN cannot carry out local training as the public graph does not contain the necessary labels for computing a local GNN. Additionally, the DP guarantees of FedGNN are achieved by adding noise directly to the local gradient of each local GNN, which is not feasible in PRIVGNN.

### E.4 Differences with SAPGNN Shan et al. (2021)

SAPGNN, like FedGNN, generates local embeddings for each data holder and updates a global model by combining all local embeddings. However, SAPGNN ensures privacy using an n-out-of-n secret sharing mechanism to update local model weights. Each data holder secretly shares their local gradient with the others, adds up the shares, and reconstructs the entire gradient by gathering the aggregated results from others. The use of a cut layer ensures that raw data is only involved in local computations and will not be accessible to others, guaranteeing data privacy. Thus, comparing PRIVGNN with SAPGNN is not feasible.

### E.5 Differences with HETERORR Tran et al. (2022)

HETERORR adopts a similar approach to the previous works above by incorporating a noise addition mechanism using randomized response to perturb the node features and graph structure directly. The original dataset is discarded, and only the noisy data is employed for training. As a result, HETERORR relies on node-level local DP and edge-level local DP guarantees. Thus, the privacy guarantees of PRIVGNN cannot be compared.

### E.6 Differences with GAP Sajadmanesh et al. (2022)

The approach employed by GAP involves obtaining lower-dimensional node features independent of the graph structure and preserving the privacy of the node features by applying DP-SGD in the case of node-level privacy, followed by the addition of Gaussian noise to the multi-hop aggregated node embeddings. During the encoding phase of GAP, node features and labels are required to compute the embedding, which differs from PRIVGNN where not all node features and labels are available. Moreover, the GAP framework also requires both the node labels, non-aggregated node embeddings (output of the encoder module), and the aggregated embeddings to train the classification module. It should be noted that the public data of PRIVGNN does not have any labels. Therefore, GAP and PRIVGNN cannot be compared due to the differences in their data requirements.

However, just to consider the possibility or to explore the idea of comparison, we can present a rough comparison as we use the same Amazon dataset (Amazon2M in PRIVGNN). According to Figure 6 (right) of the GAP paper, at an $\epsilon$ value of 16, GAP achieved 78% accuracy while PRIVGNN achieved a comparable accuracy of 70% at a similar $\epsilon$. It should be noted that the Amazon dataset used in GAP has been preprocessed to include nodes with the top 10 classes, providing high-quality data for training, whereas PRIVGNN uses the default dataset with 47 classes. Additionally, PRIVGNN used only 1000 labeled nodes (queries) for training the released model, while GAP used 1.3 million labeled nodes for training. It is essential to emphasize that this comparison is not reasonable due to the differences in the experimental settings, but it provides an idea.

### E.7 Differences with Zhang et al. (2022)

The paper employs the approximate personalized PageRank (APPR) to protect graph topology information, by either adding Gaussian noise or applying the exponential mechanism to output the top-K relevant neighbors for computing the node feature aggregation. They replace the message passing algorithm of a standard GNN model with APPR and train the modified model using DPSGD. However, their approach cannot be directly compared with PRIVGNN for several reasons. Firstly, their method of using DPSGD is not applicable to current state-of-the-art GNN models that are based on the message passing algorithm as DPSGD assumes data independence, a condition violated for graphs. Secondly, their privacy guarantee is somewhat unclear. The authors themselves cannot represent the overall epsilon, which is a combination of $\epsilon_{pr}$ and $\epsilon_{sgd}$, as a result, they kept one fixed and the other variable. To compare the two methods fairly, one could naively fix the value of K, $\epsilon_{pr}$, and $\epsilon_{sgd}$ of their approach against the epsilon at lambda=0.4 of PRIVGNN. However, such a comparison would lack a fair basis.

### E.8 Differences with Daigavane et al. (2021)

The paper, like Zhang et al. (2022) mentioned earlier, employs DPSGD to ensure privacy during GNN training. Also, their privacy guarantees are limited to 1-layer GNN, which renders a comparison infeasible. Besides, as the authors note, if an r-layer GNN is employed, then during inference for a node U, the features of all other nodes V in the entire r-layer neighborhood of U must be available non-privately. That is, to perform inference on a node U in a multi-layer GNN, one needs to access the features of all nodes in its neighborhood up to r-layers which are not achievable in our setting. More precisely, their method is not equipped to deal with the inductive scenario where there exist two separate graphs that do not intersect, which is the case in the framework of PRIVGNN.

## F Datasets

In the following, we describe the datasets used in this work.

**Amazon.** We use the Amazon Computers dataset Shchur et al. (2018) which is a subset of the Amazon co-purchase graph McAuley et al. (2015). The nodes represent products, and the node features are product reviews represented as bag-of-words. The edges indicate that two products are frequently bought together, while class labels are the product categories. We created the private graph on 2500 nodes and used 3000 nodes each for the public train and test sets. The task for this dataset is to assign products to their respective product category.

**Amazon2M.** The Amazon2M dataset Chiang et al. (2019) is the largest Amazon co-purchasing network where each node is a product, and the features are extracted via bag-of-words on the product description. Edges represent whether two products are purchased together. It consists of over 2 million nodes. Hence, Amazon2M. We created a private graph on 1 million nodes and a public graph on the remaining nodes.

**Reddit.** The Reddit dataset Hamilton et al. (2017) represents the post-to-post interactions of a user. An edge between two posts indicates that the same user commented on both posts. The labels correspond to the community of a post. We randomly sampled 300 nodes from each class for the private graph and selected 15000 nodes each for the public train and public test nodes, respectively.

**Facebook.** The Facebook dataset Traud et al. (2012) is an anonymized Facebook social network of users from 100 American institutions. We utilized the social network among UIUC students where the task is to predict their class year. The nodes represent Facebook users(students), and the edges indicate friendship. The private graph is constructed over half of the nodes and the public graph on the other half.

**ArXiv.** The ArXiv dataset Hu et al. (2020) is a citation network where each node represents an arXiv paper and edges represent that a paper cites the other. The node feature is a 128-dimensional feature vector obtained by averaging the embedding of words in the title and abstract of each paper. Our private graph is

created from 90941 papers published until 2017. We used 48603 papers published in 2018 and 29799 papers published since 2019 for our public train nodes and test nodes, respectively.

**Credit.** The Credit defaulter graph Agarwal et al. (2021) is a financial network where nodes are individuals and edges exist between individuals based on the similarity of their spending and payment pattern. The task is to determine whether an individual will default on their credit card payment. The private and public graphs consist of 10000 and 15000 nodes, respectively.

We remark that irrespective of the size of the public data, we only pseudo-label a very small number of query nodes (in our experiments, the maximum query size was 1000) of the public graph to train the student model.

Table 11: Data Statistics. $|V|$ and $|E|$ denote the number of vertices and edges in the corresponding graph dataset. $deg$ is the average degree of the graph. We select 50% of Amazon, Amazon2M, Reddit and Credit, 40% of ArXiv, and 20% of the Facebook dataset as the test set from the public graph.

| | | | Private | | | Public | | |
|---|---|---|---|---|---|---|---|---|
| | #class | $|X|$ | $|V^\dagger|$ | $|E^\dagger|$ | $deg$ | $|V|$ | $|E|$ | $deg$ |
| Amazon | 10 | 767 | 2500 | 11595 | 4.64 | 6000 | 37171 | 6.20 |
| Amazon2M | 47 | 100 | 1M | 12.7M | 12.73 | 1.4M | 21.8M | 15.08 |
| Reddit | 41 | 602 | 12300 | 266148 | 21.64 | 30000 | 876846 | 29.23 |
| Facebook | 34 | 501 | 13401 | 270992 | 20.22 | 13402 | 266141 | 24.82 |
| ArXiv | 40 | 128 | 90941 | 187419 | 2.06 | 78402 | 107900 | 1.38 |
| Credit | 2 | 13 | 10818 | 185916 | 17.19 | 15000 | 361093 | 24.07 |

# G   Baselines

**Non-Private Inductive Baseline (B1).** In the non-private inductive baseline, we train a single GNN model on all private data and test the performance of the model on the test set of the public data. This corresponds to releasing the model trained on complete private data without any privacy guarantees.

**Non-Private Transductive Baseline (B2).** This non-private baseline estimates the "best possible" performance on the public data. Specifically, a GNN model is trained using the training set (50% of the public data for Amazon, Amazon2M, Reddit, and Credit, 60% of ArXiv, and 80% of the Facebook dataset) and their corresponding ground truth labels in a transductive setting. We then test the model on the test set (the remaining 50% of the public data for Amazon, Amazon2M, Reddit, and Credit, and 40% of ArXiv and 20% of the Facebook dataset).

**PATE.** We adopt the PATE framework Papernot et al. (2016) where the private node-set is partitioned into $n$ disjoint subsets of nodes. Then we train GNN models (teacher models) on each dataset (the corresponding induced graph) separately. We query each of the teacher models with nodes from the public domain and aggregate the prediction of all teacher models based on their label counts. Then, independent random noise is added to each of the vote counts. The obtained noisy label is then used in training the student model, which is later released. We call this PateG. We also used multilayer perceptron (MLP) instead of GNN models for training on the disjoint node subset of the private graph. We denote the MLP approach as PateM. Note that only the features (no edge information) will be used for training teacher models in PateM. Our choice of $n$ was experimentally chosen. We observe that $n > 20$ for the Amazon, Facebook, Credit, and Reddit datasets and $n > 50$ for the ArXiv lead to extremely small data in each partition and low accuracy. We also set $n = 50$ for Amazon2M.

## G.1   Model and Hyperparameter Setup

For each of the private (teacher) ($\Phi^\dagger$) and public (student) ($\Phi$) GNN models, we used a two-layer GraphSAGE model Hamilton et al. (2017) with hidden dimension of 64 and ReLU activation function. We applied batch normalization to the output of the first layer. We applied a dropout of 0.5 and a learning rate of 0.01. For

the multilayer perceptron model (MLP) used in PATEM, we used three fully connected layers and ReLU activation layers. We trained all models for 200 epochs using the negative log-likelihood loss function for node classification with the Adam optimizer. All our experiments were conducted for 10 different instantiations using PyTorch Geometric library Fey & Lenssen (2019) and Python3 on 11GB GeForce GTX 1080 Ti GPU.

## H The Runtime and Space Requirement of PRIVGNN

We remark that the runtime and space requirements of our proposed PrivGnn are on par or even less than that of other private models like PATE. Importantly, since the user is only exposed to the public model, we point out that the time and space requirements of the released model are the same as that of the corresponding non-private GNN models.

For the training time, we train $Q$ models corresponding to $Q$ queries. As shown in Figure 3, we can decrease $Q$ to 500 (equivalent to training 500 teacher models). The runtime of PrivGnn can be further reduced by parallel training of models. Note that the original PATE trains a sufficiently large number of teacher models to achieve a reasonable privacy-accuracy trade-off. For instance, on the Glyph dataset (See Table 1 of Papernot et al. (2018)), PATE trained 5000 teacher models. With our requirement of a much smaller number of teacher models, PrivGnn is faster or better than PATE in terms of time and space requirements.

The space requirement of PrivGnn is the same as the space requirements of one GNN model because we need not save the trained teacher model after querying them once with the query nodes. Therefore, they can be discarded after training which leads to no significant space overhead. For PATE, on the other hand, all teacher models are required to answer each query leading to its higher space requirements.

## I Limitations and Future Works

**Query reduction.** As observed in the experiments, fewer queries significantly reduced the overall privacy budget $\epsilon$. However, this has a relative effect on the accuracy of the released model. Hence, a more sophisticated active learning approach or unsupervised methods for query reduction that can achieve a high accuracy are worth exploring.

**Analysing PrivGnn's behavior.** The use of explanation methods to analyze the properties of the learned representation of the PrivGnn is desirable. This analysis will help understand the internal workings of private models and compare the representations of private and non-private models.

