# OpenReview forum: "Releasing Graph Neural Networks with Differential Privacy Guarantees"
_TMLR — Accepted by TMLR_

### Review · Reviewer_cD6a · 2023-02-17

**Summary Of Contributions:**

This paper presents a novel method for learning Graph Neural Networks (GNN) under DP. The solution is based on PATE like framework, where the teachers are trained with private data to provide labels for previously unlabelled public data set to the student through noisy channel. The student then trains their model using the public features and the teacher provided noisy labels. However, compared to the standard PATE framework, authors suggest a subsampled method that avoids the data set splitting that is usually required for the PATE teacher training, and also improves the privacy budget through subsampling amplification. In the empirical evaluation, the method is shown to reach non-private classification accuracy with $\epsilon \approx 6$ for the Amazon2M data set. The method is shown to beat a private baseline of using the standard PATE framework.

**Audience:**

Yes

**Broader Impact Concerns:**

I don't think the paper raises broader impact concerns.

**Claims And Evidence:**

Yes

**Requested Changes:**

Recommendations and questions:
* In the "Differences with Existing Works", you mention that in prior work either the graph structure or part of the nodes are assumed to be public, and in contrast your work treats the "whole graph" as private. However, you do need the public feature set for your training right? If that is the case, I would think that this difference to earlier work should be clarified.
* Currently only the average accuracy is reported. I would suggest authors to also report the variability across the repeats to assess how robust the method is.
* Can you explain the intuition behind PrivGNN outperforming B1?
* "we conclude that the value of the hyperparameter K should be chosen based on the average degree of the graph", is the average degree of the graphs discussed somewhere? It would be nice to know those in order to evaluate this conclusion.
* Section 5.3: I find the conclusion of smaller $Q$ being better rather intuitive. However, it would be really interesting to see how small can you go before you start to get worse the privacy-utility trade-offs. So maybe you could plot Accuracy for several Q's while keeping the $\epsilon$ fixed.
* Table 3 would be much easier to interpret if you would keep the $\epsilon$ fixed between the $\gamma=0.1$ and $\gamma=0.3$ cases.

Minor things:
* Alg. 1, Step 5: The $(v)$ in subscript of $V^\dagger_{KNN}$ is inconsistent with the notation of previous step. Or maybe there is some difference that is not clear to me. It also seems to happen again in Section 3.1.1.
* "The mechanism is equivalent to the “sampling without replacement” scheme with": this is bit unclear if you don't say that $m$ is the batch size
* It seems that you have $m$ in the Input for Alg. 1, but I'm not sure if it is ever used.
* Might be worth mentioning that the GNN $\Phi^\dagger$ returns a probability vector (just so that the l1-sensitivity of 1 is clear).
* There is a weird looking line break in the 2nd paragraph of Section 5.2
* Maybe a typo: "is slightly higher with only 6% decrease"



**Strengths And Weaknesses:**

Strengths:
* The proposed method seems novel to me. While the privacy method is comprised of multiple parts, the combination seems clever to me.
* Extensive empirical evaluation for the proposed method. Authors consider several interesting research question in the experiments that provide solid ground for understanding how the method is affected by various hyperparameters.

Weaknesses:
* While the empirical evaluation covers many important questions, I think the results could be presented better. In the current form, different parameter settings tend to lead to different privacy parameters and different accuracies. Now it is really difficult to compare the setting while both of these vary. I would suggest authors to for example keep the $\epsilon$ fixed and report the accuracies obtained using different settings.

---

### Review · Reviewer_3G6U · 2023-02-17

**Summary Of Contributions:**

The authors propose a paradigm for training GNN models with differential privacy guarantee in the specific scenario where we have a private labeled graph (the teacher) and a public unlabeled graph (the student), and the goal is to release a model that can give predictions for the public graph. The main idea is to obtain noisy labels from a teacher (trained on private data) for a small number of query nodes and use them to train a public model on the public graph. To obtain DP guarantees Laplace noise is added to the teacher predictions. To amplify the privacy budget sub-sampling is used. Simple membership inference attacks are performed as a sanity check.

**Audience:**

Yes

**Broader Impact Concerns:**

There are no specific concerns that I want to raise. However, given that the paper proposes an approach for preserving privacy I would encourage the authors to further highlight the limitations of their approach, and specifically discuss what kind of privacy protection is or is not offered by their approach, as well as the limitations of DP in general. Moreover, a discussion of how to interpreted the guarantees, e.g. what a given value of $\epsilon$ means would be useful.

**Claims And Evidence:**

Yes

**Requested Changes:**

The following additional baselines might be helpful to better compare the trade-offs of the proposed model:
- Train a single GNN on the *joint* public and private graph (i.e. the original dateset), importantly, this would include the cross-edges between the public and private components of the graph
- To validate the usefulness of the teacher the authors should compare to a private baseline where Laplace noise is added directly to the ground-truth labels of the query nodes
- While some of the mentioned related works are indeed designed for a distributed setting, in principle they can be also applied (maybe with small modifications) in the same setting as PrivGNN. Are there any arguments for not comparing with them? How does the proposed approach compare to methods for computing private node embeddings?

This is a minor question, but it would be interesting to study the performance when using different models for the private GNN and public GNN.

**Strengths And Weaknesses:**

Conditional on accepting the proposed setting, the approach is well motivated. The paper is well written, well organized, and easy to follow. The authors do a good job of supporting all of their claims with solid theoretical and experimental evidence. The ablation study is well executed and convincingly shows that all introduced components are necessary. The baselines are reasonable (additional baselines can be helpful see requested changes). The addition of the membership inference attacks is appreciated.

One weakness of the approach is in the setting itself since it implicitly assumes that *all* information (features and structure) about *all* test nodes is public. That is, the privacy of the test nodes is not protected at all. This scenario seems quite narrow. For example, while it might be reasonable that a given node $v$ knows the information about its neighbors (e.g. I know information about my friends in a social network), this approach assumes that everyone in the graph knows the features and neighbors of everyone else. It would be helpful if the authors can talk about in more detail the real-world settings where their approach would be useful.

Relatedly, a second weakness of the approach is that the empirical results are likely too optimistic due to the experimental setup. Namely, the public and private graph are artificially constructed from the same original graph. In most applications it does not seem realistic that the private and public graph are essentially "the same". Can you specify a real-world scenario where roughly half of the graph is public and the other half is private? For example, in a social network, all users need to have their privacy protected.

An important oversight of the current manuscript is that it is not explicitly stated exactly what is protected with differential privacy. In general, in DP we have two neighboring inputs $X$ and $X'$ where the neighboring relation are defined depending on the context. For example $X$ and $X'$ can be two datasets that differ in one instance. In the DP+Graphs literature there are usually two notions of neighboring graphs that lead to node-level DP if $G$ and $G'$ differ in one node, or edge-level DP if they differ in one edge. The author should clearly state what is the neighboring relation that they consider. Relatedly, the use the worst possible bound on the L1 norm of the difference in predictions for the teacher. Depending on the definition of neighboring graph this can be quite pessimistic.

Minor:
- It would be helpful to add more details on the membership attack. Currently it's hard to judge the strength of the attacker.
- In the statement "In addition, the current legal data protection policies highlight a compelling  need to develop privacy-preserving GNNs" which policies are you referring to?
- In "form batches and lots", the definition of "lots" is not clear
- In paragraph 2 of section 5.2 there is an extra empty line
- While it's true that the Binomial converges to the Poisson it is not clear why this is relevant in this setting. The sampling that is performed is IID Bernoulli (or equivalently Binomial), so why is it called Poisson sampling? Does the amplification theory require Poisson sampling, and if yes what does this mean for the guarantee?

---

### Review · Reviewer_v6ag · 2023-03-30

**Summary Of Contributions:**

This paper proposed an approach to train GNN models with differential privacy based on the setting of teachers and students (PATE). The proposed approach, PrivGNN, includes three main steps: (1) Use Poisson sampling to select a random subset of nodes, which is used to induce a subgraph by gathering all k-nearest neighbors of the selected nodes. Train teacher GNNs based on the induced subgraphs and private labels; (2) Compute noisy posterior labels using the Laplace mechanism; and (3) Train student models using the noisy posterior labels. Experimental results were conducted on several datasets in comparison with several baseline models. The results show that the proposed PrivGNN achieves better accuracy – privacy trade-offs compared with the baselines. In addition, PrivGNN further reduced the performance of membership inference attacks.

**Audience:**

Yes

**Broader Impact Concerns:**

The setting is incorrect and impractical.

**Claims And Evidence:**

No

**Requested Changes:**

Please address all comments.

**Strengths And Weaknesses:**

Strengths
+ The research topic is significant.
+ The empirical results look promising.

Weaknesses
There are many significant concerns regarding the proposed approach, as follows:

- The most critical concern is the correctness of the proposed approach. The paper does not discuss the level of privacy protection offered by PrivGNN, i.e., node-level protection or edge-level protection. The paper appears to focus on node-level protection reflected through the node-level membership inference attack evaluation in Appendix C. However, that raises the question about the definition (1) neighboring induced graphs? and (2) GNN model sensitivity given neighboring induced graphs, especially given the k-nearest-neighbors of a particular node. There is no proof that the induced subgraph preserves DP. Removing one node in the induced subgraph can significantly change all other nodes' posterior (prediction). How does  PrivGNN upper-bound this sensitivity to provide DP protection? The proof is shallow and does not help justify the DP protection level (even the proofs in Appendices). For instance, if a node has many connections compared with other nodes, it will have a higher probability of appearing in more induced subgraphs given the same number of queries. In addition, there is no protection for the nodes’ features and the private labels. That can significantly affect both node-level and edge-level privacy protection of the proposed approach.

- The advantages of the proposed approach are not well justified, especially in comparison with PATE. Using only one induced subgraph to return the noisy posterior labels for a specific query should not provide better model utility than using several teacher models as in PATE. What is the missing ingredient making PrivGNN so effective?

- The setting is impractical. The privacy budget is heavily affected by the number of queries. That means an end-user can only submit a limited number of queries and must know the number of queries |Q| in advance. This is impractical in real-world applications. In addition, this is problematic since an adversary can pretend to be multiple different end-users to submit a sufficient number of queries to destroy the privacy protection offered by the proposed approach completely.

- Experimental results are unconvincing. Critical baselines are missing, which depends on the theoretical correctness of the proposed approaches (which level of privacy protection). For instance, [1-4] and more. In addition, experiments must reflect a practical setting between the service provider against the adversary, given varying factors, including the number of queries and k-nearest neighbors. Why k-nearest neighbors in comparison with other measures? Why Laplace mechanism in protecting the posterior labels? Any other methods, such as Label-DP?

[1] Towards Training Graph Neural Networks with Node-Level Differential Privacy. 2022

[2] Heterogeneous Randomized Response for Differential Privacy in Graph Neural Networks. 2022

[3] Node-level differentially private graph neural networks. 2021

[4] Gap: Differentially Private Graph Neural Networks with Aggregation Perturbation. 2022

- The presentation must be improved. Section 3.1.4 can be moved to the Appendix to reserve space for justifying the proposed approach's correctness, advantages, and novelty. A clear threat model is needed. Also, focusing on shedding light on understanding these factors rather than “reporting” the results.

---

### Decision · Action_Editors · 2023-06-06

**Recommendation:** Accept with minor revision

**Comment:**

During the discussion period, the authors provided convincing replies and a revised version of their manuscript addressing the reviewers' requests. One reviewer remained unresponsive during the discussion period, but I consider that the authors clarified the key points and appropriately addressed the raised concerns. In light of the discussion, the other two reviewers support acceptance. I believe that this work is an interesting addition to the private GNN literature, and also recommend acceptance.

That being said, an important part of the discussion revolved around the positioning and comparison to prior work on private GNNs. The authors provided some clarifications, and incorporated additional references and discussion in Appendix E. However, in the current version, this appendix is not referenced in the main text and contains imprecise statements, such as "One major difference between PrivGnn and existing works is that other methods rely on local DP for privacy guarantees, PrivGNN utilizes global DP to protect the privacy of the entire dataset", which not hold e.g. for Sajadmanesh et al. (2022). On the other hand, the related work section of the main text (Section 2) misses key references.

Therefore, the authors are asked to make a minor revision of their manuscript with the following changes:
- Incorporate the references from Appendix E into Section 2 and update the section accordingly. Detailed discussions can be kept in Appendix E, with a reference to it in Section 2.
- Update imprecise statements that are still present in Section 2 and Appendix E (such as the one mentioned above).
- Make the setting considered in this paper (public unlabeled graph + partial access to private labeled graph) more explicit in the abstract and in the intro, as this is an important element that makes the approach different from some of the prior work.



**Audience:**

This paper introduces a new approach for privacy-preserving graph neural networks, and is thus relevant to TMLR audience interested in graph-based learning on the one hand, and privacy-preserving machine learning on the other hand.

**Claims And Evidence:**

The claims of the paper are supported by matching evidence in the form of proofs of differential privacy guarantees and numerical experiments.